# Antipsychotic olanzapine-induced misfolding of proinsulin in the endoplasmic reticulum accounts for atypical development of diabetes

Satoshi Ninagawa[1]*, Seiichiro Tada[2†], Masaki Okumura[3†], Kenta Inoguchi[2†], Misaki Kinoshita[3], Shingo Kanemura[3,4], Koshi Imami[5], Hajime Umezawa[1], Tokiro Ishikawa[1], Robert B Mackin[6], Seiji Torii[7], Yasushi Ishihama[5], Kenji Inaba[8], Takayuki Anazawa[2], Takahiko Nagamine[9], Kazutoshi Mori[1]*

[1]Department of Biophysics, Graduate School of Science, Kyoto University, Kyoto, Japan; [2]Department of Surgery, Graduate School of Medicine, Kyoto University, Kyoto, Japan; [3]Frontier Research Institute for Interdisciplinary Sciences, Tohoku University, Sendai, Japan; [4]School of Science and Technology, Kwansei Gakuin University, Sanda, Japan; [5]Department of Molecular and Cellular BioAnalysis, Graduate School of Pharmaceutical Sciences, Kyoto University, Kyoto, Japan; [6]Department of Biomedical Sciences, Creighton University School of Medicine, Omaha, United States; [7]Laboratory of Secretion Biology, Institute for Molecular and Cellular Regulation, Gunma University, Maebashi, Japan; [8]Institute of Multidisciplinary Research for Advanced Materials, Tohoku University, Sendai, Japan; [9]Sunlight Brain Research Center, Yamaguchi, Japan

*For correspondence:
sninagawa@upr.biophys.kyoto-u.ac.jp (SN);
mori@upr.biophys.kyoto-u.ac.jp (KM)

†These authors contributed equally to this work

Competing interests: The authors declare that no competing interests exist.

**Abstract** Second-generation antipsychotics are widely used to medicate patients with schizophrenia, but may cause metabolic side effects such as diabetes, which has been considered to result from obesity-associated insulin resistance. Olanzapine is particularly well known for this effect. However, clinical studies have suggested that olanzapine-induced hyperglycemia in certain patients cannot be explained by such a generalized mechanism. Here, we focused on the effects of olanzapine on insulin biosynthesis and secretion by mouse insulinoma MIN6 cells. Olanzapine reduced maturation of proinsulin, and thereby inhibited secretion of insulin; and specifically shifted the primary localization of proinsulin from insulin granules to the endoplasmic reticulum. This was due to olanzapine's impairment of proper disulfide bond formation in proinsulin, although direct targets of olanzapine remain undetermined. Olanzapine-induced proinsulin misfolding and subsequent decrease also occurred at the mouse level. This mechanism of olanzapine-induced β-cell dysfunction should be considered, together with weight gain, when patients are administered olanzapine.

## Introduction

Patients with schizophrenia are typically prescribed first- or second-generation antipsychotics (FGAs or SGAs, respectively). FGAs block dopamine signaling in the brain by inhibiting the function of its receptors, resulting in recovery from conditions such as acute mania and agitation (*Divac et al., 2014*). However, because of excessive inhibition of the dopamine pathway, FGAs can cause extrapyramidal symptoms, including dystonia, akathisia, and Parkinsonism, and are now widely replaced by SGAs due to their reduced risk of causing adverse extrapyramidal effects (*Høiberg and Nielsen,*

2006) and greater effectiveness in alleviating symptoms (*Lee et al., 2002*; *Sirota et al., 2006*). These effects of SGAs are caused by their inhibition of signaling through the dopamine $D_2$ receptor and 5-$HT_{2A}$ serotonin receptor (*Divac et al., 2014*; *Kasper et al., 1999*). Of note, SGAs have higher risks of causing metabolically adverse side effects such as obesity (*Gothelf et al., 2002*), dyslipidemia (*Gothelf et al., 2002*) and diabetes mellitus (*Citrome and Volavka, 2005*; *Koller and Doraiswamy, 2002*).

The SGA olanzapine affects neurotransmitter receptors (called multi-acting receptor-targeted antipsychotics) and thereby exhibits significantly greater effectiveness in alleviating various symptoms than other SGAs (*Sirota et al., 2006*; *van Bruggen et al., 2003*). However, it carries a higher risk of diabetes than other SGAs such as risperidone, and its frequent use therefore represents a clinical problem (*Deng, 2013*; *Newcomer, 2005*; *Rummel-Kluge et al., 2010*). Generally, olanzapine-induced diabetes is attributed to weight gain caused by increased appetite through the effect on the feeding center via blocking of 5-$HT_{2C}$ receptor, leading to insulin resistance-mediated development of diabetes.

Nonetheless, olanzapine-induced diabetes includes atypical cases, as follows. First, while it usually takes years to develop diabetes via insulin resistance, olanzapine-induced diabetes occurs within 6 months after commencing treatment (*Kinoshita et al., 2014*; *Nakamura et al., 2014*; *Nakamura and Nagamine, 2010*). Second, discontinuing olanzapine treatment cures diabetes even after the level of HbA1c (a marker for blood glucose level) reaches >10%, a ratio which normally representing an irreversible level in patients with type I or type II diabetes (*Nakamura et al., 2014*; *Nakamura and Nagamine, 2010*; *Nathan et al., 2009*; *Young et al., 2012*). Moreover, diabetic ketoacidosis, which is caused by insulin hyposecretion, often occurs in patients with type I diabetes, but also rapidly (within 6 months after treatment with olanzapine) affects patients with no diabetic symptoms before medication (*Kinoshita et al., 2014*; *Tsuchiyama et al., 2004*). Indeed, epidemiological data show that patients receiving olanzapine incur an approximately 10-times higher risk of diabetic ketoacidosis than the general population (*Polcwiartek et al., 2016*), suggesting the possibility that olanzapine directly impairs the function of pancreatic β cells, the exclusive source of insulin, in a speculated few percent of medicated patients (*Nagamine, 2014*; *Nagamine, 2018*).

We are interested in the mechanism of the development of olanzapine-induced atypical diabetes from the viewpoint of the protein quality control system operating in the endoplasmic reticulum (ER). Secretory and membrane proteins gain their tertiary and quaternary structures in the ER, which is assisted by ER-localized molecular chaperones, glycosyltransferases and oxidoreductases (collectively referred to here as ER chaperones). Only correctly folded molecules are transported from the ER to the Golgi apparatus and then to their destination. However, proteins that fail to fold correctly, even with the assistance of ER chaperones, are retrotranslocated into the cytosol and degraded by the ubiquitin-proteasome system (*Qi et al., 2017*). This disposal system is called ER-associated degradation (ERAD).

Although eukaryotic cells perform quality control of proteins through ER chaperone-assisted productive folding and ERAD, the requirements for protein synthesis may exceed the protein folding capacity of the ER under physiological and pathological conditions. To counteract such ER stress, namely the accumulation of unfolded or misfolded proteins in the ER, the unfolded protein response (UPR) is triggered, leading to the activation of three ER transmembrane sensor proteins PERK, ATF6, and IRE1. Interestingly, their activation effects come out sequentially. First, phosphorylation of the α subunit of eukaryotic translation initiation factor 2 (eIF2α) by activated PERK transiently suppresses translation (see *Figure 1A*) so that the burden of the ER is mitigated because of reduced translocation of newly synthesized proteins into the ER. Next, activated ATF6 upregulates the transcription of ER chaperone genes to increase the folding capacity of the ER. Subsequently, activated IRE1 together with activated ATF6 upregulates the transcription of ERAD component genes, which increases degradation capacity. If ER stress is prolonged, apoptotic signaling is activated (*Mori, 2000*; *Tabas and Ron, 2011*; *Yamamoto et al., 2007*; *Yoshida et al., 2003*).

Pancreatic β cells synthesize preproinsulin, fold proinsulin in the ER, and store large quantities of insulin in insulin granules (see *Figure 1A*). In these roles, they are frequently subject to ER stress, and the UPR accordingly plays a critical role in maintaining homeostasis. Immediate translational control mediated by the PERK pathway, rather than time-consuming transcriptional control of ER chaperones etc. (*Yoshida et al., 2003*), is particularly important for the folding of proinsulin, a very small molecule compared with ER chaperones; when translation is generally attenuated by the

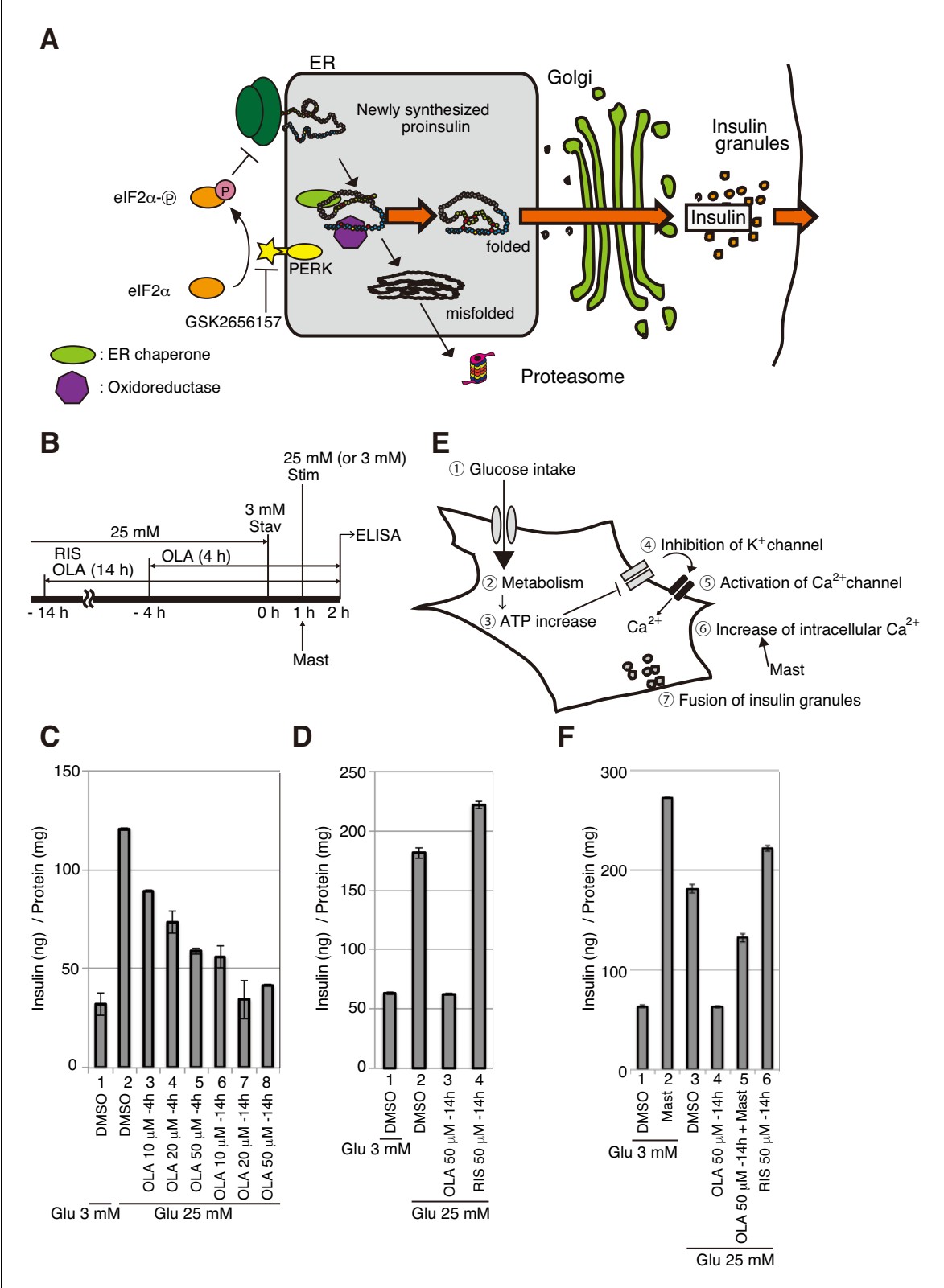

**Figure 1.** Effect of olanzapine on secretion of insulin from MIN6 cells. (**A**) Schematic representation of protein quality control in the ER, PERK-mediated translational attenuation, and storage and secretion of insulin (see Introduction). GSK2656157 inhibits protein kinase activity of PERK. (**B**) Scheme for measuring the level of insulin secreted into medium in response to glucose stimulation. After culture in a medium containing 25 mM glucose, MIN6 cells were starved for 1 hr in medium containing 3 mM glucose (Stav), and then cultured in medium containing 25 mM glucose (Stim or 3 mM glucose

*Figure 1 continued on next page*

*Figure 1 continued*

as control). The amount of insulin secreted into the medium during the 1 hr incubation was determined using an ELISA. The data are normalized to the amounts of total cellular proteins and presented as the mean ± SD (n = 2). (C) MIN6 cells were pretreated with the indicated concentrations of olanzapine (OLA) for 4 hr or 14 hr in medium containing 25 mM glucose before glucose starvation as shown in (B). (D) MIN6 cells were pretreated with 50 µM olanzapine or risperidone (RIS) for 14 hr before glucose starvation as shown in (B). (E) Schematic representation of the signaling cascade for insulin secretion in response to glucose stimulation. Cells intake glucose (①). Glucose metabolism (②) triggers an increase in cellular ATP/ADP ratio (③), which inhibits $K^+$ channels and induces depolarization (④). This leads to opening of $Ca^{2+}$ channels (⑤) and increased influx of $Ca^{2+}$ (⑥), inducing the fusion of insulin granules to the plasma membrane for secretion (⑦). Mastoparan (Mast) enhances insulin secretion by increasing the intracellular concentration of $Ca^{2+}$ independently of glucose intake. (F) MIN6 cells were pretreated with 50 µM olanzapine or risperidone for 14 hr before glucose starvation as shown in (B). Mastoparan (Mast, 20 µM) was added at the time of glucose stimulation to aliquots of cells not pretreated or pretreated with 50 µM olanzapine for 14 hr.

activation of PERK, stochastically unfolded or misfolded proinsulin can be efficiently refolded by the action of preexisting ER chaperones. If PERK is not active, synthesis of misfolded proinsulin continues, leading to proteotoxicity-mediated apoptosis of β cells. Accordingly, PERK knockout mice develop diabetes after birth (*Harding et al., 2001*). Further, PERK mutation causes human Wolcott-Rallison syndrome, which induces diabetes in infancy (*Delépine et al., 2000*). We previously found that olanzapine induces mild ER stress in hamster pancreatic β-cell line HIT-T15, which secretes insulin. However, phosphorylation of eIF2α is not elevated despite mild activation of PERK in olanzapine-treated HIT-T15 cells, leading to sustained protein synthesis followed by induction of apoptosis, although we used olanzapine at the concentration of 100 µM (*Ozasa et al., 2013*).

We also previously found that olanzapine markedly inhibits insulin secretion by HIT-T15 cells (*Ozasa et al., 2013*). In the present study, we confirmed that 10–50 µM olanzapine inhibits insulin secretion by the mouse pancreatic β-cell line MIN6, which is frequently used to study β cells. We then investigated the stage of protein quality control at which insulin secretion is blocked in olanzapine-treated MIN6 cells, and then whether the identified stage is indeed compromised in islets of mice after daily oral administration of olanzapine.

## Results

### Inhibition of insulin secretion in olanzapine-treated MIN6 cells

MIN6 cells exhibit an approximately 7-fold increase in insulin secretion in 2 hr when the extracellular glucose concentration is increased from 5 to 25 mM (*Ishihara et al., 1993*). Here, we detected an approximately 4-fold increase in insulin secretion in 1 hr when the extracellular glucose concentration was increased from 3 to 25 mM (stimulation after starvation for 1 hr, *Figure 1B*) compared with continuous incubation in 3 mM glucose (*Figure 1C*, compare bar 2 with bar 1). Glucose-stimulated insulin secretion was inhibited in a concentration-dependent manner when MIN6 cells were first treated with 25 mM glucose in the presence of 10, 20, or 50 µM olanzapine for 4 hr (*Figure 1C*, bars 3–5) and was markedly inhibited by pretreatment for 14 hr with these same concentrations of olanzapine (*Figure 1C*, bars 6–8), but not by pretreatment for 14 hr with 50 µM risperidone (*Figure 1D*, compare bar 4 with bar 2). Pretreatment was essential for this effect, because the addition of 50 µM olanzapine from the beginning of the 1 hr starvation did not detectably inhibit glucose-stimulated insulin secretion (data not shown), suggesting that it takes time for olanzapine to exert its inhibitory effect on insulin secretion.

Glycolysis causes ATP-mediated closure of the potassium ($K^+$) channels, leading to opened calcium ($Ca^{2+}$) channel-mediated influx of extracellular $Ca^{2+}$. The increase in cytosolic $Ca^{2+}$ concentrations culminates in fusion of insulin granules with the plasma membrane and subsequent secretion of insulin (*Figure 1E*; *Rorsman and Braun, 2013*). Therefore, insulin secretion can be induced by mastoparan, which increases intracellular $Ca^{2+}$ concentrations by activating phospholipase C to produce inositol 1, 4, 5-triphosphate (*Perianin and Snyderman, 1989*). Indeed, treatment of MIN6 cells with mastoparan for 1 hr after glucose starvation rapidly stimulated insulin secretion (*Figure 1F*, compare bar 2 with bar 1). Pretreatment of MIN6 cells with 50 µM olanzapine for 14 hr partially inhibited mastoparan-induced insulin secretion (*Figure 1F*, compare bar 5 with bar 2). These results suggest that olanzapine may not block $Ca^{2+}$-induced fusion of insulin granules with plasma membrane, but may decrease the amount of insulin stored in insulin granules.

## Retention of proinsulin in the ER in olanzapine-treated MIN6 cells

Proinsulin produced from preproinsulin by the action of signal peptidase becomes mature proinsulin via formation of three intramolecular disulfide bonds in the ER, and is then processed to insulin (*Figure 2A*). To analyze MIN6 cells which secrete both proinsulin and insulin (*Lee et al., 2011*; *Tsuchiya et al., 2018*), as is seen when isolated mouse and human islets were analyzed by pulse-chase experiments (*Dufurrena et al., 2019*), we employed two types of mouse monoclonal antibodies with different characteristics, namely #8138 raised against a synthetic peptide corresponding to the residues surrounding Val36 of human insulin, which are conserved in mouse insulin (yellow circle denotes Val36 in *Figure 2A*), and I2018 raised against human insulin. Under reducing conditions, #8138 detected mainly proinsulin in lysates of MIN6 cells, and both proinsulin and insulin B chain in lysates of mouse islets, probably reflecting their relative contents (*Figure 2B*). It should be noted that #8138 was 800-fold more reactive with purified recombinant human proinsulin than purified recombinant human insulin (insulin B chain) under reducing conditions (*Figure 2C*), and that #8138 reacted with neither proinsulin nor insulin under non-reducing conditions (*Figure 2B*).

A pulse (20 min)-chase (80 min) experiment showed that the immunoprecipitates obtained from MIN6 cells using #8138 were detected as doublet bands which migrated slightly faster than the 8.5 kDa marker and were secreted into medium within the 80 min chase (*Figure 3A and B*). On comparison of the migration positions of in vitro translational products, the upper band was considered to be proinsulin 2 (*Figure 3B*), consistent with the abundant expression of insulin 2 mRNA in MIN6 cells (*Roderigo-Milne et al., 2002*). The lower band was considered to be processed proinsulin 2, such as the B chain connected to the C peptide but not to the A chain as observed in isolated rat islets (*Harding et al., 2012*), or somewhat cleaved proinsulin, because only proinsulin 2 was detected after a shorter pulse (3 min) and processed proinsulin 2 appeared after the 30 min chase (*Figure 3C*). Processed proinsulin 2 is likely to be produced after moving out from the ER, because its level was markedly decreased by treatment with brefeldin A, an inhibitor of anterograde transport from the ER to the Golgi apparatus (*Figure 3D*). #8138 immunoprecipitated proinsulin 2 and processed proinsulin 2 but not insulin 2 from MIN6 cells after pulse (20 min)-chase (reducing conditions, *Figure 3E*).

In contrast, I2018 detected insulin in lysates of MIN6 cells and mouse islets only under non-reducing conditions (*Figure 2B*). A pulse (20 min)-chase experiment showed that I2018 immunoprecipitated both proinsulin 2 and insulin 2 from MIN6 cells (reducing conditions, *Figure 3E*). This also demonstrated the precursor-product relationship for proinsulin 2 and insulin 2 in MIN6 cells. Importantly, I2018 immunoprecipitated mature proinsulin 2, which folded into a compact structure and thereby migrated slightly faster than proinsulin 2 immunoprecipitated using #8138, in addition to insulin 2 (non-reducing conditions, *Figure 3E*). Hereafter, #8138 and I2018 were appropriately used in accordance with experimental purposes.

We examined the effect of olanzapine and risperidone on the synthesis and secretion of proinsulin and insulin in MIN6 cells by pulse-chase experiments (*Figure 4A*). Analysis under reducing conditions of immunoprecipitates obtained with #8138 showed that production of intracellular processed proinsulin (P′) as well as secretion of both proinsulin (P) and processed proinsulin (P′) were markedly decreased by treatment of MIN6 cells with 50 µM olanzapine (*Figure 4B*, lanes 8–14) but not with 50 µM risperidone (lanes 29–35) compared with control cells (lanes 1–7), and that proinsulin (P) was instead accumulated intracellularly in olanzapine-treated MIN6 cells [see quantified *Figure 4B* (a)]. These phenomena were observed when MIN6 cells were similarly treated with 10 µM olanzapine but not with 10 µM risperidone (*Figure 5A and B*). Analysis under non-reducing conditions of immunoprecipitates obtained with I2018 showed that the levels of both intracellular mature proinsulin (mP) and insulin (I) were markedly decreased and that secretion of both mature proinsulin (mP) and insulin (I) was markedly inhibited by treatment of MIN6 cells with 50 µM olanzapine (*Figure 4B*, lanes 22–28) but not with 50 µM risperidone (lanes 36–42) compared with control cells (lanes 15–21) [see quantified *Figure 4B* (b)], consistent with the ELISA data (*Figure 1*). Thus, olanzapine blocked maturation of proinsulin and accordingly inhibited secretion of insulin.

Phogrin, a transmembrane protein-tyrosine phosphatase-like protein, transits the secretory pathway (*Torii et al., 2005*), and furin-like convertases convert proinsulin and phogrin to their mature forms by cleaving them at a dibasic consensus site (*Hermel et al., 1999*; *Nakayama, 1997*). Pulse-chase experiments showed that olanzapine did not detectably affect the processing of transfected phogrin-GFP (*Figure 6A*) or secretion of transfected α1-proteinase inhibitor (A1PI) (*Figure 6B*). The

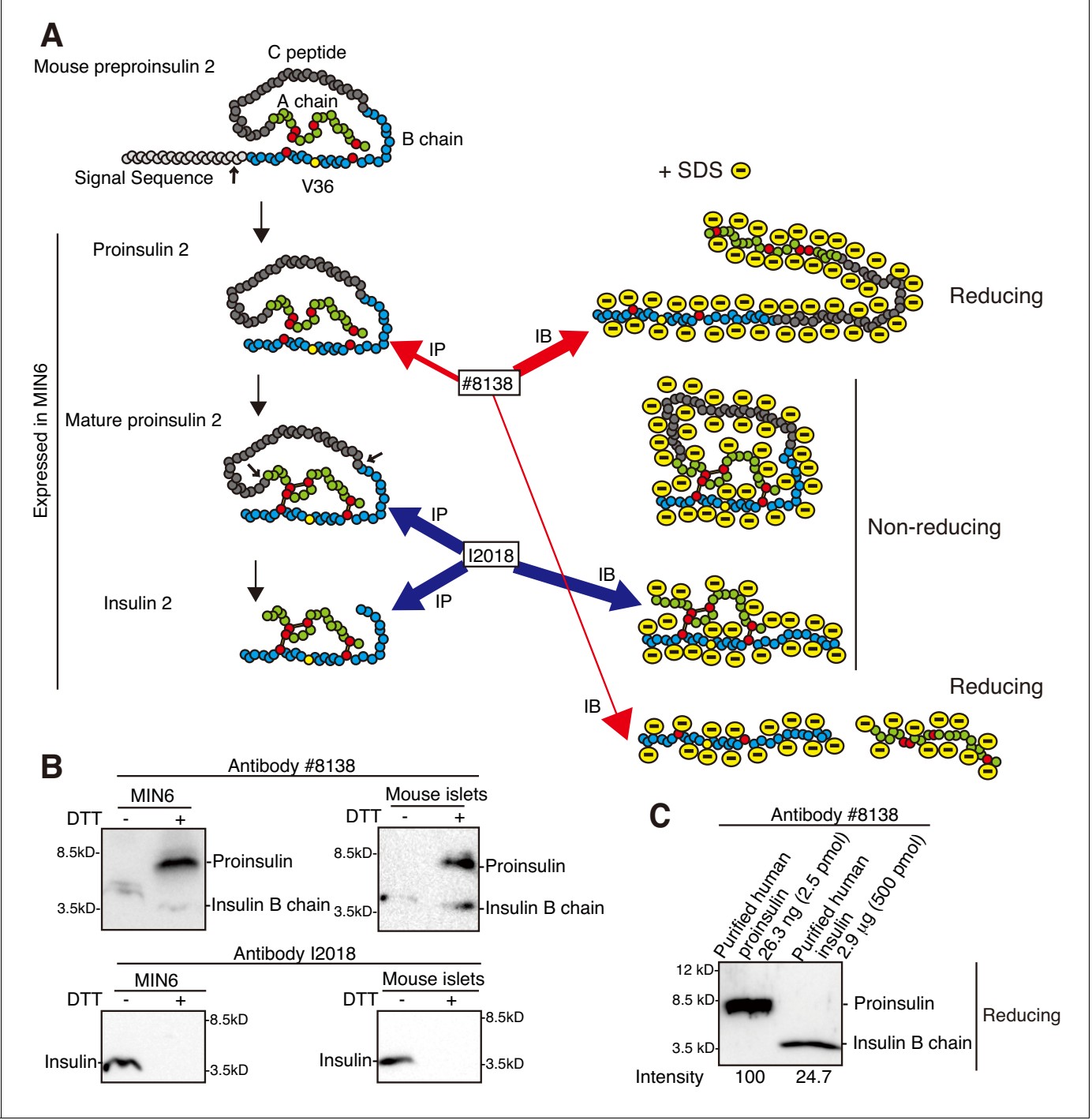

**Figure 2.** Characterization of two anti-insulin monoclonal antibodies utilized. (**A**) Schematic representation of maturation of insulin 2. Each circle denotes an amino acid residue. Preproinsulin 2 comprises a signal sequence (light gray), B chain (blue), C peptide (gray), and A chain (green). Preproinsulin 2 is converted to proinsulin 2 after cleavage of the signal sequence, and proinsulin 2 becomes mature proinsulin 2 via formation of three intramolecular disulfide bonds. Mature proinsulin 2 is converted to insulin 2 after proteolysis at the residues indicated by the two arrows. The red circles represent cysteine residues that form the disulfide bonds (bars), and which are required for insulin folding and activity. The region encompassing V36, indicated by the yellow circle, is recognized by the anti-insulin monoclonal antibody #8138, which immunoprecipitates (immature) proinsulin 2 and detects proinsulin 2 and insulin 2 (B chain) after reducing SDS-PAGE. In contrast, the anti-insulin monoclonal antibody I2018 immunoprecipitates mature proinsulin 2 and insulin 2, and detects insulin 2 after non-reducing SDS-PAGE. (**B**) Lysates of MIN6 cells and mouse islets were analyzed by immunoblotting using #8138 and I2018 after reducing (DTT +) and non-reducing (DTT -) SDS-PAGE. (**C**) The indicated amounts of recombinant and purified proinsulin and insulin were subjected to reducing SDS-PAGE followed by immunoblotting using #8138.

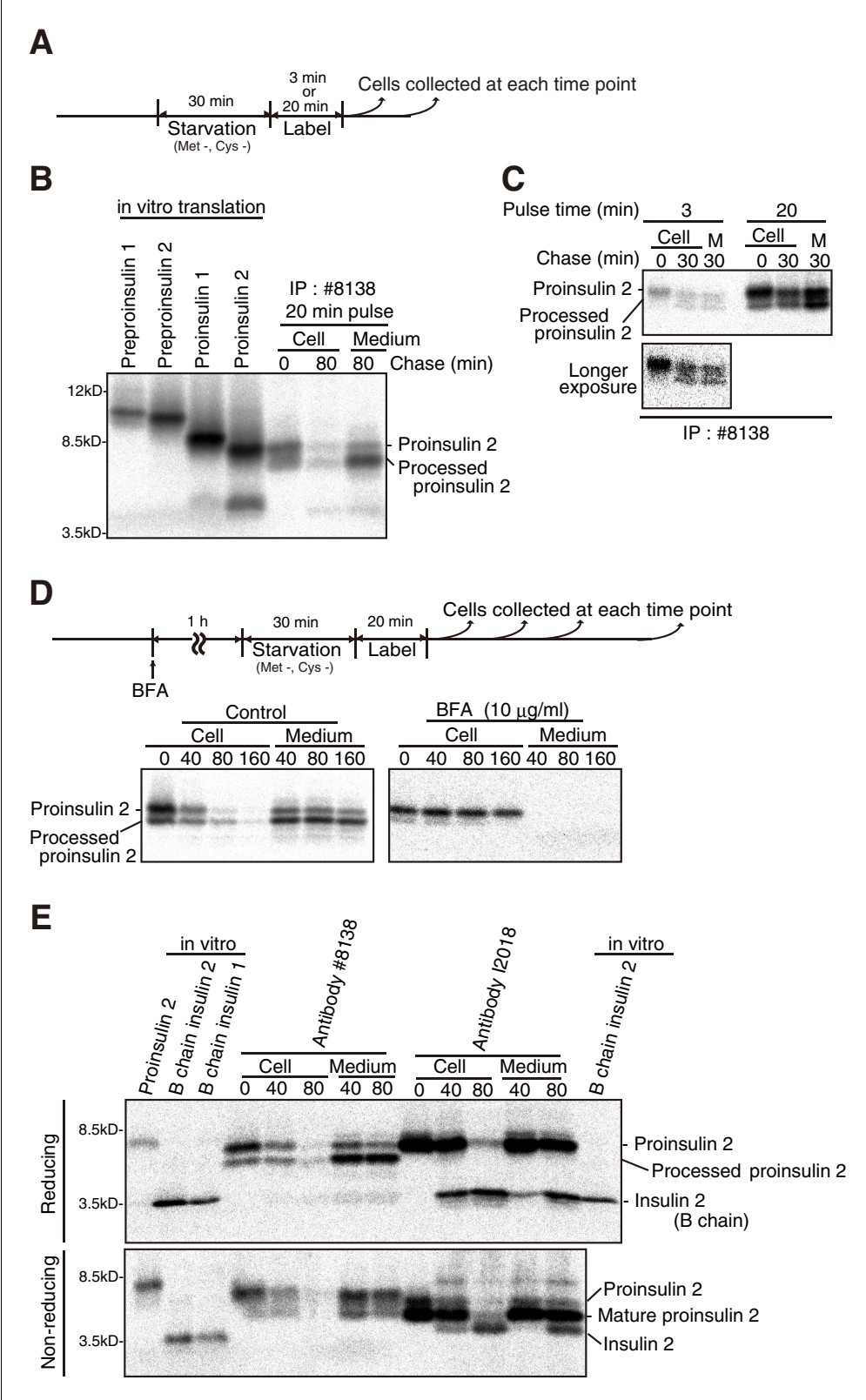

**Figure 3.** Secretion of proinsulin 2, processed proinsulin 2 and insulin 2 from MIN6 cells. (**A**) Schemes of the experiments shown in (**B**) and (**C**). (**B**) cDNAs encoding mouse preproinsulin 1, preproinsulin 2, proinsulin 1, and proinsulin 2 were subjected to in vitro transcription and translation in the presence of $^{35}$S-Met and $^{35}$S-Cys, and then to immunoprecipitation using #8138. A pulse (20 min)-chase (80 min) experiment was performed in MIN6 cells, and cell lysates and media were collected and subjected to immunoprecipitation using #8138. The immunoprecipitates were analyzed by

*Figure 3 continued on next page*

*Figure 3 continued*

reducing SDS-PAGE and autoradiography. (**C**) A pulse (3 or 20 min)-chase (30 min) experiment was performed in MIN6 cells, and cell lysates and media (M) were analyzed as in (**B**). (**D**) MIN6 cells untreated or treated with brefeldin A (BFA, 10 µg/ml) were subjected to pulse-chase experiment to determine changes in the levels of intracellular and extracellular proinsulin as well as processed proinsulin, as shown in the schema (top). (**E**) cDNAs encoding mouse proinsulin 2, B chain of mouse insulin 1, and B chain of mouse insulin 2 were treated as in (**B**). A pulse (20 min)-chase (40 min and 80 min) experiment was performed in MIN6 cells, and cell lysates and media were collected and subjected to immunoprecipitation using #8138 or I2018. The immunoprecipitates were analyzed by reducing and non-reducing SDS-PAGE followed by autoradiography.

anti-KDEL antibody stained the ER by recognizing the major ER chaperones BiP/GRP78 and GRP94, and A1PI was widely distributed from the ER to the Golgi apparatus before and after olanzapine treatment (*Figure 6C and D*). Olanzapine also did not significantly affect the maturation of hemagglutinin from high-mannose type to complex type (*Figure 6E*), which requires correct disulfide bond formation for folding (*Segal et al., 1992*). These results confirmed the specificity of olanzapine's effect to proinsulin.

Immunofluorescence analysis using #8138 revealed that proinsulin mainly localized to perinuclear compartments, which were distinct from the ER reacted with the anti-calnexin antibody (*Figure 7A*) but overlapped with the localization of phogrin-GFP (*Figure 7B*), suggesting that they were included in immature or mature secretory granules.

We were surprised to find that proinsulin colocalized with calnexin in MIN6 cells treated with 50 µM olanzapine (*Figure 7C*) and that this shift in localization was detected in MIN6 cells treated with 10 µM olanzapine (*Figure 7D*) but not in MIN6 cells treated with 50 µM or 10 µM risperidone (*Figure 7E and F*). In contrast, the localization of phogrin-GFP (*Figure 7G and H*) or insulin (*Figure 8A and B*) was not altered by treatment with 50 µM olanzapine. These results suggest that olanzapine specifically affects the quality of proinsulin during its localization in the ER.

## Induction of proinsulin misfolding in olanzapine-treated MIN6 cells

Treatment of MIN6 cells with the proteasome inhibitor MG132 alone increased the levels of intracellular and extracellular proinsulin (*Figure 5C and D*, left panels), indicating that a part of newly synthesized proinsulin is constitutively subjected to ERAD in MIN6 cells. Importantly, simultaneous treatment of MIN6 cells with olanzapine and MG132 markedly increased the level of intracellular proinsulin (*Figure 5D*, right top panel), indicating that proinsulin forced to be remain in the ER of olanzapine-treated cells was subjected to ERAD. Nonetheless, ERAD of retained proinsulin alone cannot explain olanzapine-induced blockage of proinsulin secretion, because proinsulin secretion was still blocked in MIN6 cells treated with both olanzapine and MG132 (*Figure 5D*, right bottom panel), in contrary to the case with control cells (*Figure 5D*, left bottom panel).

As ERAD deals with misfolded proteins, we examined the effect of olanzapine on the solubility of proinsulin by lysing MIN6 cells with 1% NP40 followed by centrifugation, and found that a small amount of proinsulin became insoluble in MIN6 cells after treatment with 10 µM and 50 µM olanzapine but not with 50 µM risperidone (*Figure 8C*). We next checked whether olanzapine affects the disulfide-bonded status of proinsulin using #8138, which unlike I2018 does not recognize mature proinsulin (see *Figure 3E*). By analyzing under non-reducing conditions of immunoprecipitates obtained with #8138 after pulse chase (*Figure 4A*), we found that an approximately 27 kDa form of proinsulin, designated as a high molecular weight form of proinsulin (HMP-1), and an approximately 15 kDa form of proinsulin, designated as HMP-2, were produced in cells treated with olanzapine (*Figure 4B*, lanes 8–14) but not with risperidone (lanes 29–35) compared with control cells (lanes 1–7) [see quantified *Figure 4B* (c)]. HMP-1 and HMP-2 detected with #8138 but not with I2018 were not secreted at all, suggesting that they were severely misfolded.

To identify the components of HMP-1 and HMP-2, we conducted mass spectrometric analysis. To this end, olanzapine-untreated or -treated cells were lysed in buffer containing 10 mM N-ethylmaleimide (NEM) to maintain existing disulfide bonds. Subsequent non-reducing SDS-PAGE and immunoblotting revealed that proinsulin and HMP-2 somehow became undetected with #8138 (*Figure 8D*). Therefore, we were able to purify only HMP-1 by immunoprecipitation (*Figure 8E*). Mass

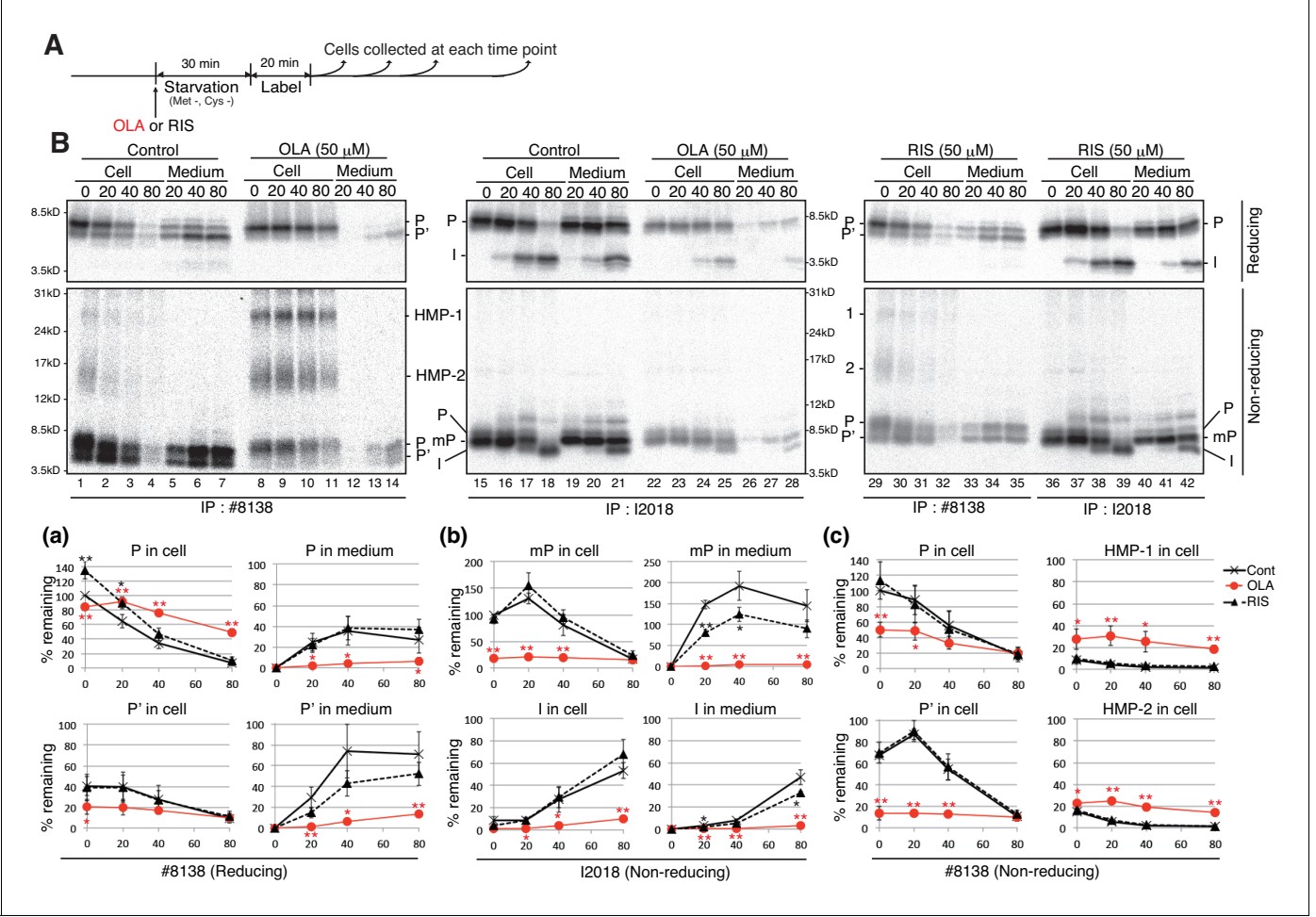

**Figure 4.** Effect of olanzapine and risperidone on maturation and secretion of proinsulin and insulin in MIN6 cells. (**A**) Schemes of the experiments shown in (**B**). (**B**) MIN6 cells untreated or treated with olanzapine (50 µM) or risperidone (50 µM) from the start of starvation were subjected to pulse-chase experiment to determine changes in the levels of proinsulin (P), processed proinsulin (P'), mature prinsulin (mP), insulin (I), HMP-1 (1) and HMP-2 (2) in cells and medium. Cell lysates and media were collected and immunoprecipitated with #8138 or I2018. The immunoprecipitates were analyzed by reducing and non-reducing SDS-PAGE followed by autoradiography. The intensity of each band was determined, and the intensity of intracellular proinsulin or mature proinsulin at time 0 in control cells was defined as 100% (n = 3) [categorized in **a**, **b** and **c**].

spectrometric analysis of HMP-1 revealed that only insulin 2-derived fragments were enriched upon olanzapine treatment (*Figure 8F and G*, and *Supplementary file 1*), suggesting that HMP-2 and HMP-1 represent a proinsulin dimer and trimer, respectively, with aberrant intermolecular disulfide bonds.

To clarify how olanzapine induces the formation of aberrant proinsulin oligomers, we considered the possibility that olanzapine acts on certain oxidoreductases in the ER to inhibit their activities (*Jang et al., 2019*; *Okumura et al., 2014*), and therefore tested whether olanzapine directly binds to the purified enzymes using isothermal titration calorimetry (ITC), which sensitively detects heat generation or absorption upon ligand-substrate binding. However, results were negative for PDI, ERp46, ERp57, ERp72, and P5 (*Figure 8—figure supplement 1A*). We were also unable to precisely analyze direct interactions between olanzapine and proinsulin by ITC, because reduced proinsulin (*Figure 8—figure supplement 1B*) was rapidly and severely aggregated, producing significant heat exchange. Thus, direct targets of olanzapine remain to be determined.

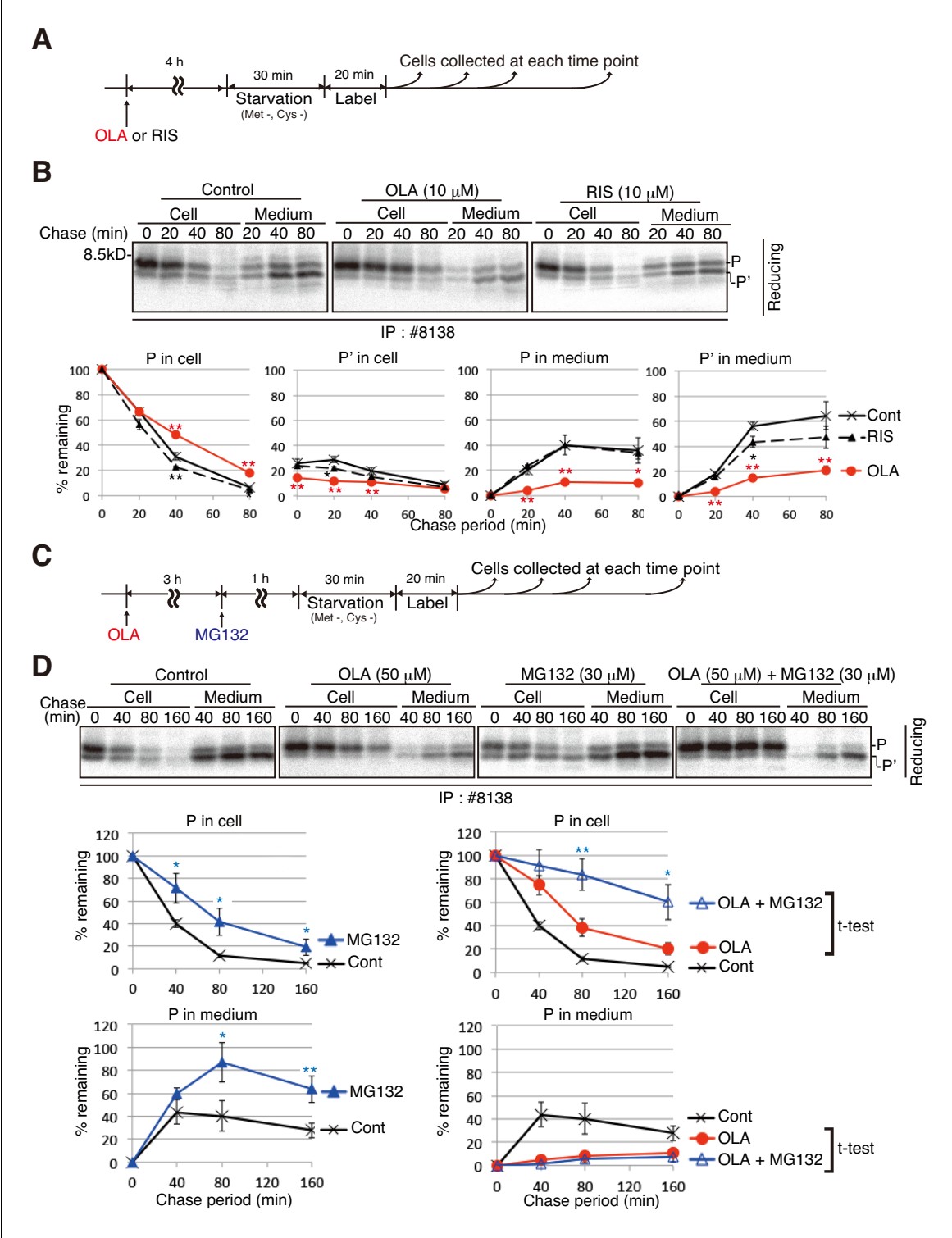

**Figure 5.** Effect of the proteasome inhibitor MG132 on the level of proinsulin accumulated intracellularly in olanzapine-treated MIN6 cells. (**A**) Schemes of the experiments shown in (**B**). (**B**) MIN6 cells untreated or treated with olanzapine (10 μM) or risperidone (10 μM) for 4 hr were analyzed as in *Figure 4B* to determine changes in the levels of proinsulin (P) and processed proinsulin (P') in cells and medium using #8138 and reducing SDS-PAGE. The intensity of intracellular proinsulin at time 0 was defined as 100% (n = 3). (**C**) Schemes of the experiments shown in (**D**). (**D**) MIN6 cells untreated or treated with olanzapine (50 μM), MG132 (30 μM) or olanzapine (50 μM) and MG132 (30 μM) were analyzed as in *Figure 4B* to determine changes in the level of proinsulin (P) in cells and medium using #8138 and reducing SDS-PAGE. The intensity of intracellular proinsulin at time 0 was defined as 100% (n = 3).

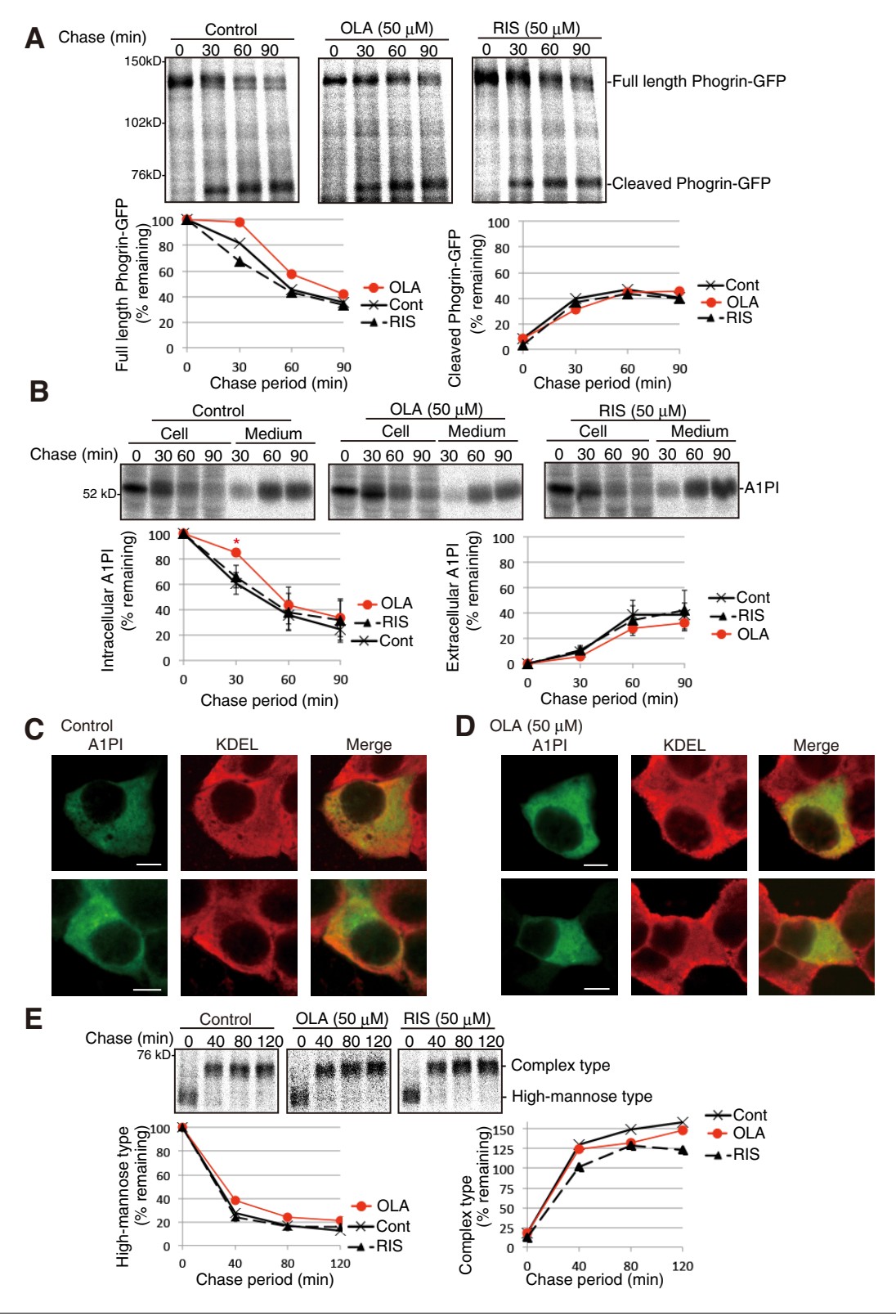

**Figure 6.** Effect of olanzapine on processing of phogrin-GFP, secretion of A1PI, and maturation of hemagglutinin in MIN6 cells. (**A**) MIN6 cells transfected with a phogrin-GFP expression vector were untreated or treated with olanzapine (50 μM) or risperidone (50 μM) for 4 hr and then subjected to pulse-chase experiment in the presence of olanzapine (50 μM) or risperidone (50 μM) to determine the rate of processing of phogrin-GFP. The amounts of full-length and cleaved phogrin-GFP were determined and are shown below with the amount of full-length phogrin-GFP at time 0 defined

*Figure 6 continued on next page*

*Figure 6 continued*

as 100% (n = 1). (**B**) MIN6 cells transfected with an A1PI expression vector were untreated or treated with olanzapine (50 μM) or risperidone (50 μM) for 4 hr and then subjected to pulse-chase experiment in the presence of olanzapine (50 μM) or risperidone (50 μM) to determine the rate of secretion of A1PI. The amounts of intracellular and extracellular A1PI were determined and are shown below with the amount of intracellular A1AP at time 0, defined as 100% (n = 3). (**C**) (**D**) MIN6 cells transfected with an A1PI expression vector were untreated (**C**) or treated with olanzapine (50 μM) for 14 hr (**D**). Fixed and permeabilized cells were analyzed by immunofluorescence using anti-A1PI and anti-KDEL antibodies. Bars: 5 μm. (**E**) MIN6 cells transfected with a hemagglutinin expression vector were untreated or treated with olanzapine (50 μM) or risperidone (50 μM) for 4 hr and then subjected to pulse-chase experiment to determine the rate of maturation of hemagglutinin. The immunoprecipitates were digested with endoglycosidae H and then analyzed by reducing SDS-PAGE and autoradiography. The amounts of high-mannose type and complex type hemagglutinin were determined and are shown below, with the amounts of high-mannose type hemagglutinin at time 0 defined as 100% (n = 1).

## ERAD of misfolded proinsulin in olanzapine-treated MIN6 cells and mouse islets

To determine whether olanzapine treatment indeed induces the production of aberrant high molecular weight forms of proinsulin and how quickly it does so, MIN6 cells pulse-labeled for 20 min were chased with the addition of olanzapine (*Figure 9A*). Analysis under non-reducing conditions of immunoprecipitates obtained with #8138 showed that most radioactivity was recovered as proinsulin in untreated cells (0 min), and secretion of proinsulin was inhibited in cells treated with olanzapine, as expected (*Figure 9B*). Importantly, the levels of HMP-1 and HMP-2 rapidly and markedly increased after treatment with olanzapine (*Figure 9B*).

To correlate the production of HMP-1 and HMP-2 with the ERAD of proinsulin, we added olanzapine together with or without MG132 at the start of starvation and then conducted pulse-chase experiments (*Figure 9C*), because we found that treatment with olanzapine produced maximal amounts of HMP-1 and HMP-2 within 40 min (*Figure 9B*). Analysis under reducing conditions of immunoprecipitates obtained with #8138 showed that the results obtained in *Figure 9D* (upper part) were consistent with those shown in *Figure 5D* for the level of intracellular and extracellular proinsulin. Critically, analysis under non-reducing conditions of immunoprecipitates obtained with #8138 revealed that MG132 stabilized proinsulin monomer, HMP-1, and HMP-2 in olanzapine-treated MIN6 cells (*Figure 9D*, lower part). These results strongly suggest that proinsulin monomer, HMP-1, and HMP-2 with aberrant disulfide bonds underwent ERAD (see *Figure 10G*).

We finally examined whether olanzapine induces misfolding of proinsulin at the mouse level. Non-reducing SDS-PAGE analysis followed by immunoblotting showed that HMP-1 and even higher molecular weight forms of proinsulin were produced when isolated mouse islets were incubated for 4 h with 50 μM olanzapine (*Figure 10A*), for 20 h with 5 μM and 10 μM olanzapine (*Figure 10B*) but not with 10 μM risperidone (*Figure 10C*) without affecting the molecular mass of PDI, an abundant ER-resident oxidoreductase (*Figure 10B*). Reducing SDS-PAGE analysis followed by immunoblotting showed that the level of proinsulin was decreased 4 h after incubation with 50 μM olanzapine (*Figure 10A*) and that this decrease was blocked by the addition of 30 μM MG132 from 1 h earlier (total 5 h) (*Figure 10D*). Of note, the level of proinsulin was markedly decreased 20 h after incubation with 5 μM and 10 μM olanzapine (*Figure 10B*).

Furthermore, non-reducing SDS-PAGE analysis followed by immunoblotting showed that HMP-1 and even higher molecular weight forms of proinsulin started to be detected in mouse islets isolated 1 week after daily oral administration of 3 mg/kg/day and 10 mg/kg/day olanzapine (*Figure 10E*), and were clearly detected in mouse islets isolated 2 weeks after daily oral administration of 10 mg/kg/day olanzapine (*Figure 10E*), and 5 weeks (*Figure 10F*) and 6 weeks (data not shown) after daily oral administration of 3 mg/kg/day olanzapine. Reducing SDS-PAGE analysis followed by immunoblotting showed that the level of proinsulin was decreased in mouse islets 5 weeks after daily oral administration of 3 mg/kg/day olanzapine (*Figure 10F*), although there was no significant change in the levels of blood glucose or plasma insulin at this stage (data not shown). These phenomena were not due to obesity because we observed no significant difference in body weight between control mice and mice after daily oral administration of 3 mg/kg/day olanzapine for 5 or 6 weeks (data not shown), but rather due to olanzapine-induced misfolding of proinsulin in pancreatic β cells.

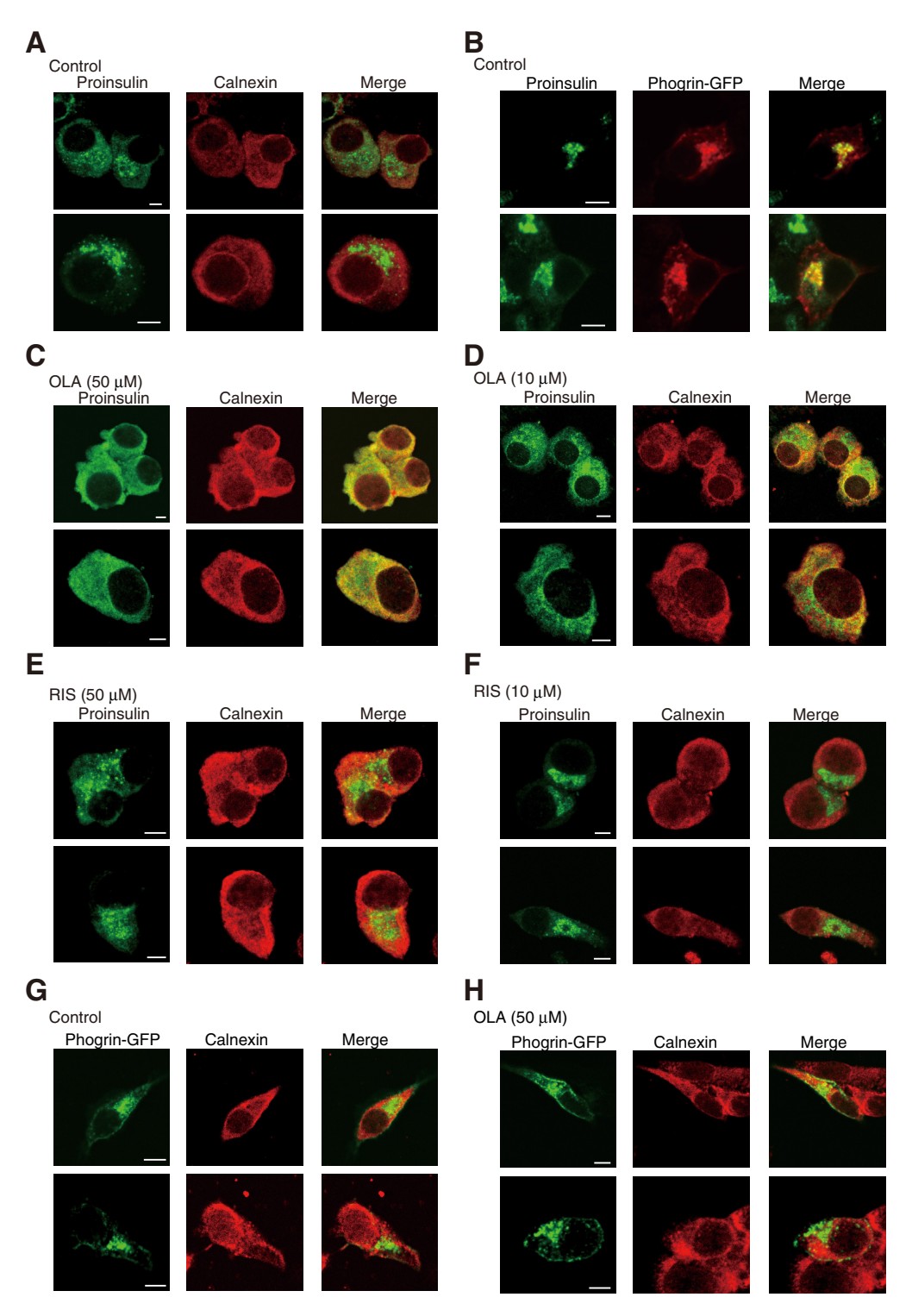

**Figure 7.** Effect of olanzapine on localization of proinsulin and phogrin-GFP in MIN6 cells (Bars: 5 µm). (**A**) MIN6 cells were analyzed by immunofluorescence using anti-insulin #8138 and anti-calnexin antibodies. (**B**) MIN6 cells transfected with a phogrin-GFP expression vector were analyzed by immunofluorescence using anti-insulin #8138 and anti-GFP antibodies. (**C**) – (**F**) MIN6 cells treated with (**C**) olanzapine (50 µM), (**D**) olanzapine (10 µM), (**E**) risperidone (50 µM), or (**F**) risperidone (10 µM) for 14 hr were analyzed by immunofluorescence using anti-insulin #8138 and anti-calnexin antibodies. (**G**) (**H**) MIN6 cells transfected with a phogrin-GFP expression vector were (**G**) untreated or (**H**) treated with olanzapine (50 µM) for 14 hr, and then analyzed by immunofluorescence using anti-GFP and anti-calnexin antibodies.

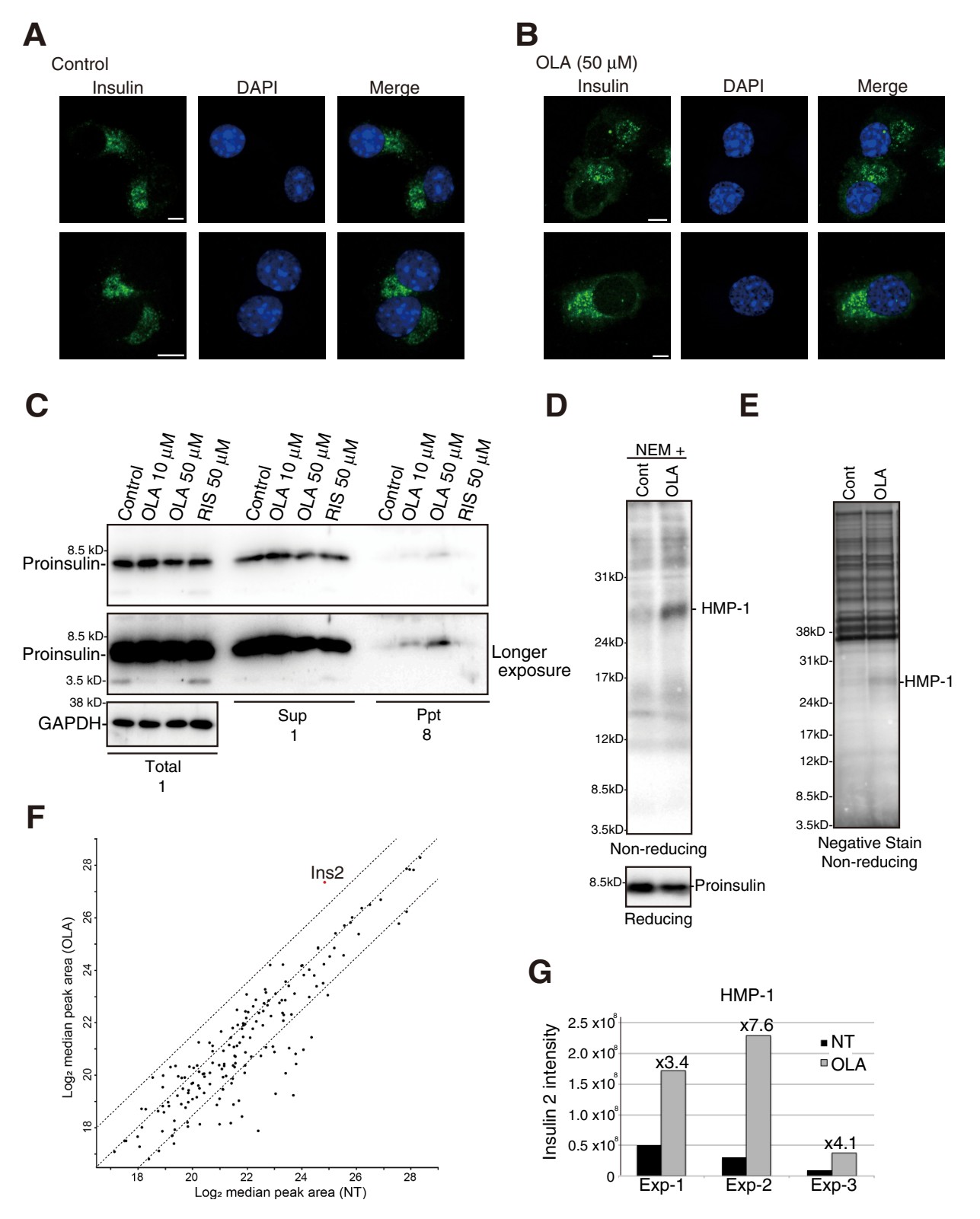

**Figure 8.** Effect of olanzapine on localization of insulin as well as solubility and oligomerization of proinsulin in MIN6 cells. MIN6 cells untreated (**A**) or treated with olanzapine (50 μM) (**B**) for 14 hr were analyzed by immunofluorescence using anti-insulin I2018 antibody. Bars: 5 μm. (**C**) MIN6 cells treated with DMSO (control), olanzapine (10 or 50 μM) or risperidone (50 μM) for 14 hr were lysed in 1% NP40. After centrifugation at 14,000 rpm for 10 min, supernatant and precipitate were analyzed by immunoblotting using anti-insulin #8138 and anti-GAPDH antibodies. Eight times greater amounts were

*Figure 8 continued on next page*

*Figure 8 continued*

used to analyze precipitate than total and supernatant. (**D**)-(**G**) MIN6 cells untreated or treated with olanzapine (50 μM) for 4 hr were lysed with 1% NP40 buffer containing 10 mM NEM. (**D**) Cell lysates were analyzed by reducing and non-reducing SDS-PAGE followed by immunoblotting using #8138. (**E**) Cell lysates were subjected to immunoprecipitation using #8138, and then to negative staining after non-reducing SDS-PAGE. (**F**) Gels at the position of HMP-1 were excised and analyzed by mass spectrometry. The results are shown by the scatter plot of log2 of the median peak area from three independent experiments between untreated cells (X axis) and olanzapine-treated cells (Y axis). A 5.7-fold increase by olanzapine treatment was observed for Ins2 as shown in the red circle. (**G**) Intensities of Ins2-derived fragments in untreated and olanzapine-treated cells in each experiment are shown along with the fold-induction.

The online version of this article includes the following figure supplement(s) for figure 8:

**Figure supplement 1.** Interaction of olanzapine with various oxidoreductases.

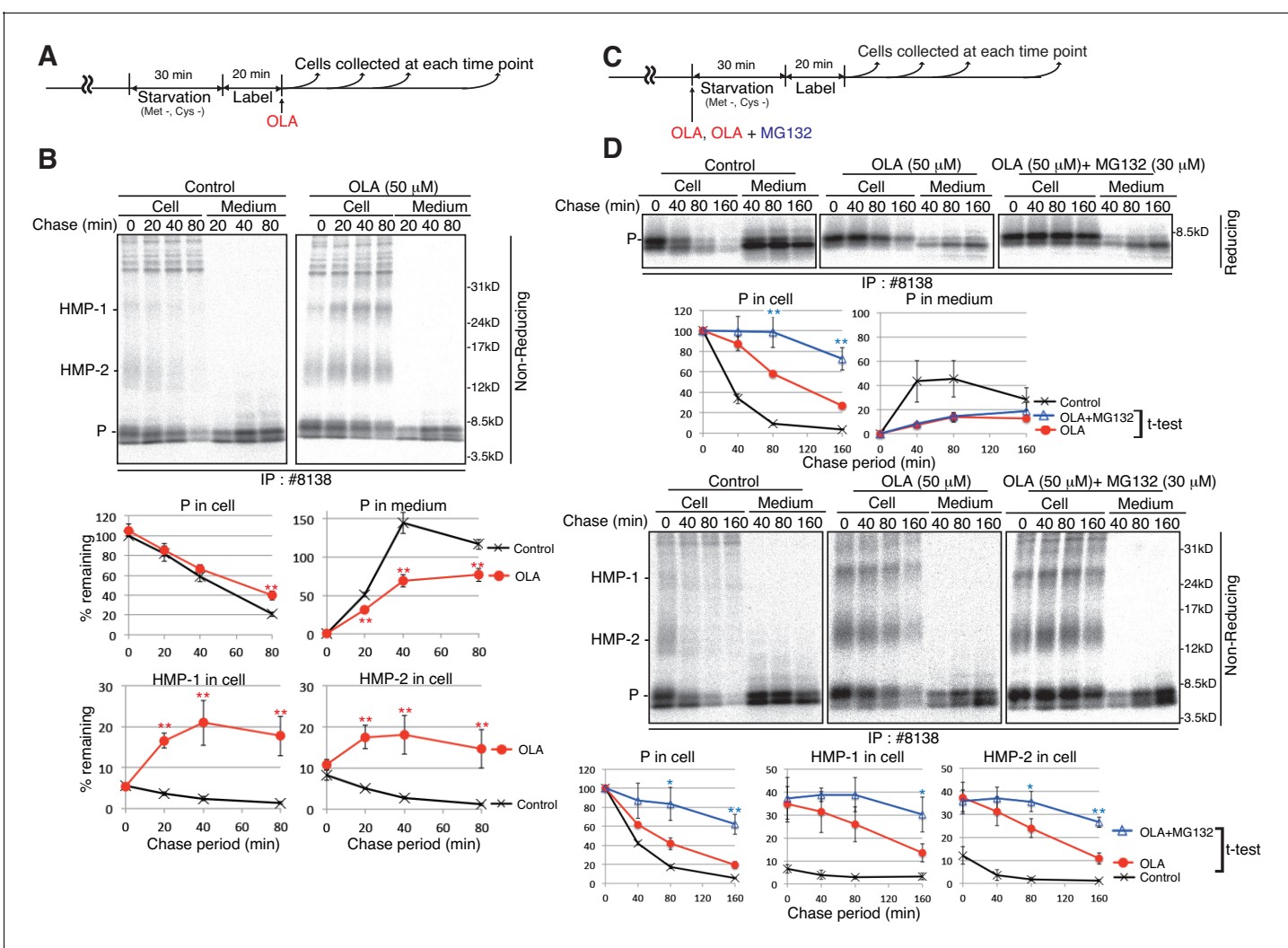

**Figure 9.** Effect of the proteasome inhibitor MG132 on the levels of proinsulin, HMP-1, and HMP-2 in olanzapine-treated MIN6 cells. (**A**) Schemes of the experiments shown in (**B**). (**B**) MIN6 cells pulse-labeled for 20 min were chased for the indicated periods with or without 50 μM olanzapine to determine changes in the levels of proinsulin (P), HMP-1 (1), and HMP-2 (2) in cells and medium as in *Figure 4B* using #8138 and non-reducing SDS-PAGE. The intensity of intracellular proinsulin at time 0 was defined as 100% (n = 3). (**C**) Schemes of the experiments shown in (**D**). (**D**) MIN6 cells untreated, treated with olanzapine (50 μM), or olanzapine (50 μM) and MG132 (30 μM) were analyzed as in *Figure 4B* to determine changes in the levels of proinsulin (P), HMP-1 (1), and HMP-2 (2) in cells and medium using #8138 and reducing and non-reducing SDS-PAGE. The intensity of intracellular proinsulin at time 0 was defined as 100% (n = 3).

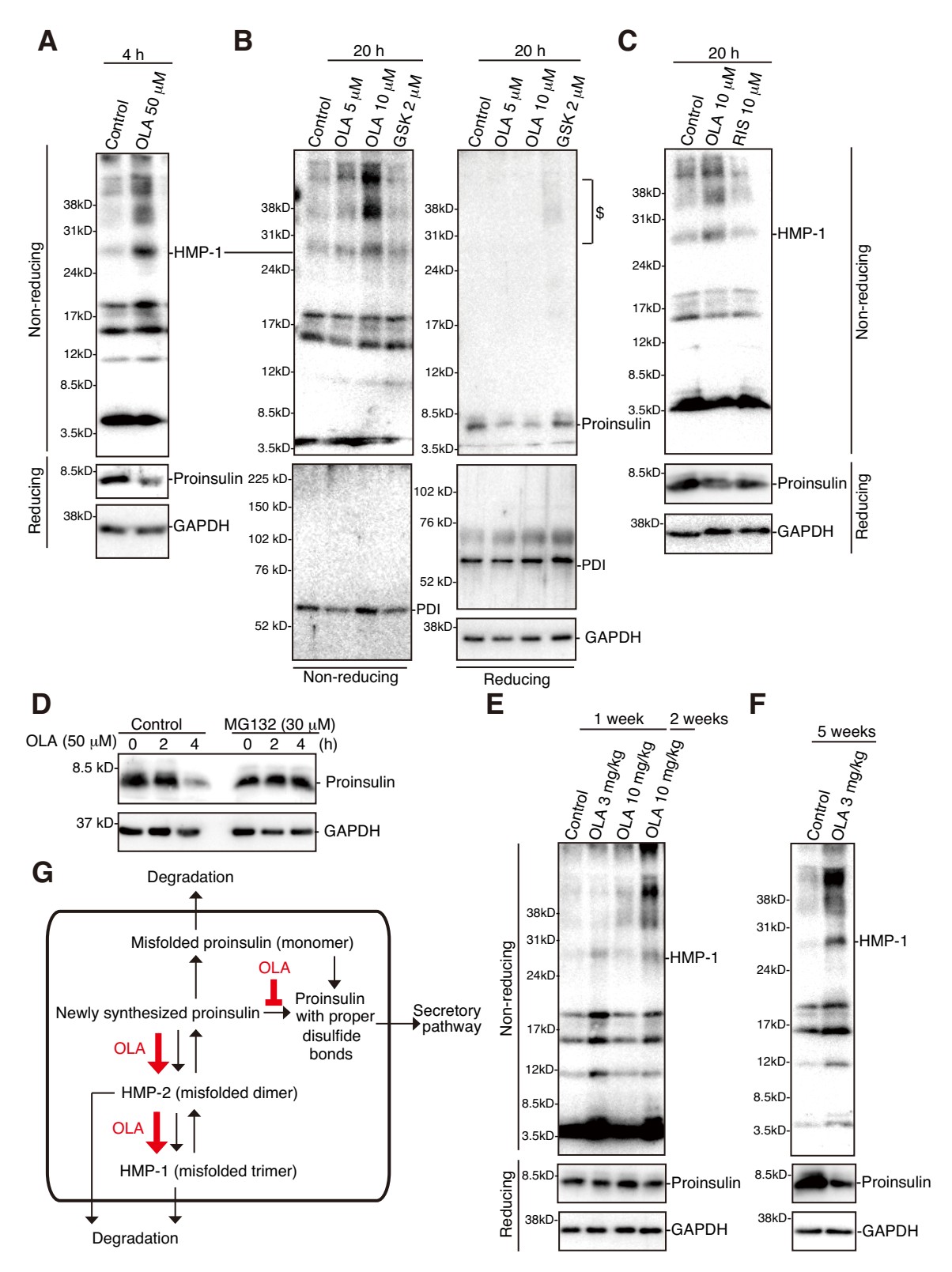

**Figure 10.** Effect of olanzapine on proinsulin oligomerization in mouse islets. (**A**)-(**C**) Isolated mouse islets were untreated or treated with the indicated concentration of olanzapine, GSK2656157 (GSK, 2 µM) or risperidone (10 µM) for the indicated period and analyzed by immunoblotting using anti-insulin #8138, anti-PDI and anti-GAPDH antibodies after reducing and non-reducing SDS-PAGE. $ denotes aggregated proinsulin. (**D**) Isolated mouse islets were treated or untreated with MG132 (30 µM) for 1 hr, then treated with olanzapine (50 µM) for the indicated period, and analyzed by

*Figure 10 continued on next page*

*Figure 10 continued*

immunoblotting using anti-insulin #8138 and anti-GAPDH antibodies after reducing SDS-PAGE. (E) (F) Islets were isolated from mice the indicated week after daily oral administration of the indicated dose of olanzapine, and analyzed by immunoblotting using anti-insulin #8138 and anti-GAPDH antibodies after reducing and non-reducing SDS-PAGE. (G) Model for olanzapine-induced β-cell dysfunction (see text).

## Discussion

Here we identified a novel mechanism that accounts for olanzapine-induced atypical development of diabetes. For this purpose, we used the mouse pancreatic β−cell line MIN6 to show that olanzapine produced aberrantly disulfide-bonded proinsulin (*Figure 4*) and caused retention of misfolded proinsulin in the ER (*Figure 7*) which then underwent ERAD (*Figure 9*). Inhibition of insulin secretion as well as proinsulin secretion (*Figures 1C* and *5B*) and retention of misfolded proinsulin in the ER (*Figure 7D*) was observed in MIN6 cells treated with 10 μM olanzapine, which is comparable with concentrations of olanzapine in patients sera (0.016–0.24 μM) and in postmortem serum (0.032–16 μM) (*Robertson and McMullin, 2000*), as well as with the observation that olanzapine concentrations in tissues are 4- to 46-fold higher than those in the plasma of rats (*Aravagiri et al., 1999*). It should be noted that olanzapine-induced apoptosis was not previously observed in MIN6 cells or in the hamster pancreatic β−cell line HIT-T15 treated with 10 μM olanzapine (*Ozasa et al., 2013*), indicating that olanzapine-induced misfolding of proinsulin is an early event which evokes ER stress in pancreatic β cells.

Importantly, olanzapine-induced misfolding of proinsulin, which is targeted to ERAD, also occurred when mouse islets were treated with olanzapine (*Figure 10A–D*). Furthermore, misfolding of proinsulin was induced in mouse islets after daily oral administration of 3 mg/kg/day olanzapine (*Figure 10E and F*). Patients usually take 10–20 mg olanzapine a day (0.17–0.33 mg/kg assuming 60 kg as body weight), which may correspond to 2.0–4.0 mg/kg in mice, if the difference in body surface areas is considered (multiply by 12.3) (*Nair and Jacob, 2016*). This mechanism can explain why certain patients rapidly develop diabetes as well as diabetic ketoacidosis after receiving olanzapine, without gaining weight, and why such patients recover when olanzapine is discontinued.

In this connection, Peter Arvan and colleagues recently showed that aberrantly disulfide-bonded proinsulin dimers and higher ladder complexes (trimer, tetramer, pentamer, —) exist in rat pancreatic β-cell-derived INS1E cells as well as in mouse and human islets, and that the levels of these misfolded forms of proinsulin were markedly increased and instead the level of insulin was markedly decreased in islets of leptin receptor mutant LepR$^{db/db}$ mice, a mouse model of diabetes, compared with islets of wild-type mice, before the onset of diabetes. They therefore proposed that proinsulin misfolding is an early event in the progression to type 2 diabetes (*Arunagiri et al., 2019*). The levels of these misfolded forms of proinsulin were dramatically increased after treatment of INS1E cells and islets of mouse and human with 2 μM GSK2656157, a potent PERK inhibitor (*Atkins et al., 2013*), for 20 hr and overnight, respectively, but not for 5 hr (*Arunagiri et al., 2019*), supporting the importance of the PERK branch of the UPR for maintenance of homeostasis of β cells, as described in the Introduction.

We confirmed the much more profound misfolding of proinsulin after treatment of mouse islets for 20 hr with 2 μM GSK2656157 than with 10 μM olanzapine, which was observed even under reducing conditions (*Figure 10B*; depicted by $, and data not shown). Interestingly, however, production of HMPs was observed after treatment of mouse islets for 20 hr with 10 μM olanzapine but not with 2 μM GSK2656157 (*Figure 10B*), and a ladder of HMPs bigger than HMP-1 was observed more clearly in mouse islets (*Figure 10*) than in MIN6 cells (*Figure 9*). This rapidity and broad extensiveness of olanzapine's action on the folding status of proinsulin in mouse islets suggests that β-cell dysfunction may infrequently dominate and diabetes might develop without weight gain or insulin resistance in the minority of patients with potentially lower catabolism of olanzapine or by other unidentified factors. For example, it is known that olanzapine is catabolized by the action of cytochrome p450 (CPY) 1A2 and polymorphic CPY2D6 in the liver and that the induction of CPY1A2 by smoking significantly diminishes the concentration of olanzapine in plasma (*Carrillo et al., 2003*); as hospitalized patients cannot smoke, the concentration of olanzapine may be elevated in plasma and possibly in β cells of ex-smokers. The clearance of olanzapine also appears to vary by race, sex and age (*Meibohm et al., 2002*).

Correct formation of three intramolecular disulfide bonds is essential for insulin folding and activity (*Haataja et al., 2016*; *Chang et al., 2003*), which may be difficult even in unstressed cells. A part of newly synthesized proinsulin is therefore constitutively degraded by the proteasome (*Figure 5D*). We show that treatment with olanzapine markedly inhibited maturation and secretion of proinsulin, and instead induced aberrant disulfide bond-formation during the folding of proinsulin, leading to the formation of HMP-1 and HMP-2 (*Figure 4*), although we could not detect direct interaction between olanzapine and oxidoreductases in the ER by ITC (*Figure 8—figure supplement 1*).

Proinsulin in Akita mice with the *Ins2* C96Y mutation fails to form the correct disulfide bonds, and is degraded by ERAD (*He et al., 2015*; *Ron, 2002*). To avoid unnecessary degradation of correctly folded proinsulin, the ER nucleotide exchange factor Grp170 distinguishes misfolded proinsulin from correctly structured proinsulin for disposal (*Cunningham et al., 2017*). Similarly, olanzapine-induced HMP-1 and HMP-2 may be recognized by Grp170 for targeting to ERAD as well. Proinsulin is not a glycoprotein and is degraded via the non-glycoprotein ERAD pathway, which can process severely misfolded glycoproteins as well (*Ninagawa et al., 2015*).

In conclusion, the mechanism identified here that mediates olanzapine-induced β-cell dysfunction should be considered, along with weight gain, in mitigating adverse side effects when patients with schizophrenia are prescribed olanzapine.

# Materials and methods

## Key resources table

| Reagent type (species) or resource | Designation | Source or reference | Identifiers | Additional information |
|---|---|---|---|---|
| Cell line (*Mus musculus*) | Insulinoma | *Miyazaki et al., 1990* | MIN6 | The cell line has been authenticated and tested negative for mycoplasma. |
| Recombinant DNA reagent | p3xFlag-CMV-14 | Sigma-Aldrich | | |
| Recombinant DNA reagent | pcDNA3.1(+) | ThermoFisher | | |
| Recombinant DNA reagent | Phogrin-GFP | *Saito et al., 2011* | | |
| Recombinant DNA reagent | A1PI | *Ninagawa et al., 2015* | | |
| Recombinant DNA reagent | HA | *Gething and Sambrook, 1982* | | |
| Antibody | Anti-insulin (Mouse monoclonal) | Cell signaling | Cat#: #8138 | WB (1:1000), IP (1:400), IF (1:100) |
| Antibody | Anti-insulin (Mouse monoclonal) | Sigma-Aldrich | Cat#: I2018 | WB (1:1000), IP (1:400) |
| Antibody | Anti-GFP (Mouse monoclonal) | Roche | Cat#: 11814460001 | IP (1:400), IF (1:100) |
| Antibody | Anti-calnexin (Rabbit polyclonal) | Enzo Life Sciences | Cat#: ADI-SPA-865 | WB (1:1000), IF (1:100) |
| Antibody | Anti-PDI (Rabbit polyclonal) | Enzo Life Sciences | Cat#: ADI-SPA-890 | WB (1:1000) |
| Antibody | Anti-A1AT (Rabbit polyclonal) | Dako | Cat#: A0012 | IP (1:400), IF (1:100) |

*Continued on next page*

*Continued*

| Reagent type (species) or resource | Designation | Source or reference | Identifiers | Additional information |
|---|---|---|---|---|
| Antibody | Anti-GAPDH (Rabbit polyclonal) | Trevigen | Cat#: 2275-PC-100 | WB (1:1000) |
| Antibody | Anti-HA (Rabbit polyclonal) | Recenttec | Cat#: R4-TP1411100 | IP (1:400) |
| Antibody | Anti-KDEL (Mouse monoclonal) | MBL | Cat#: M181-3 | IF (1:1000) |

## Statistics

Statistical analysis was conducted using Student's t-test, with probability expressed as *p<0.05 and **p<0.01 for all figures.

## Cell culture and transfection

MIN6 cells (*Miyazaki et al., 1990*) were cultured in Dulbecco's modified Eagle's medium (DMEM containing 4.5 g/liter glucose) supplemented with 2 mM L-glutamine, 10% fetal bovine serum, and antibiotics (50 U/ml penicillin and 50 µg/ml streptomycin) at 37°C in a humidified atmosphere containing 5% $CO_2$/95% air. Transfection was performed as described previously (*Ninagawa et al., 2015*) using Polyethyleneimine max (Polysciences) to replace Lipofectamine 2000 (Invitrogen). MIN6 cells were transfected 1 day after seeding, and 2 days later, pulse-chase experiments or immunofluorescence analysis were conducted.

## Reagents and antibodies

MG132 from the Peptide Institute; mastoparan from Wako; Z-VAD-fmk from Promega; and DMEM with or without 4.5 g/l (25 mM) glucose and protease inhibitor cocktail from Nacalai Tesque. Olanzapine and risperidone were purchased from Toronto Research Chemicals. Various antibodies were obtained as described in Key Resource Table.

## Construction of plasmids

Recombinant DNA techniques were performed according to standard procedures (*Sambrook et al., 1989*) and the integrity of all constructed plasmids was confirmed by extensive sequencing analyses. Mouse insulin cDNA was amplified from total RNA isolated from MIN6 cells using the primer pairs described in *Supplementary file 2*. These amplified fragments were inserted between the EcoRV and XhoI sites of pcDNA3.1 (Invitrogen). The plasmid to express mPhogrin-GFP (*Saito et al., 2011*), A1AT (*Ninagawa et al., 2015*) or hemagglutinin (*Gething and Sambrook, 1982*) was previously described.

## ELISA

Approximately $3 \times 10^5$ of MIN6 cells were plated in 24-well plates and cultured in DMEM containing 25 mM glucose for 3 days before pretreatment for 14 hr with olanzapine or risperidone, or for 4 days before pretreatment for 4 hr with olanzapine or risperidone. Cells were washed twice with Krebs-Ringer Bicarbonate (KRB) buffer (10 mM HEPES, approximately pH 7.0, containing 120 mM NaCl, 4.7 mM KCl, 2.5 mM $CaCl_2$, 1.2 mM $MgCl_2$, 1.2 mM $KH_2PO_4$, 25 mM $NaHCO_3$, and 0.1% BSA) with 3 mM glucose (*Minami et al., 2000*). Cells were starved for glucose for 1 hr in KRB buffer with 3 mM glucose containing olanzapine or risperidone, washed twice with KRB buffer with 3 mM glucose, and stimulated for glucose for 1 hr in KRB buffer with 25 mM glucose containing olanzapine or risperidone, with or without mastoparan. The buffer was collected, and the cells were lysed with buffer A (50 mM Tris/HCl, pH 8.0, containing 1% NP-40, 150 mM NaCl, protease inhibitor cocktail, 20 µM MG132, and 2 µM Z-VAD-fmk). After brief centrifugation, the protein concentrations of the lysates were determined using the BCA protein assay reagent kit (Pierce). Insulin content was

determined using an ELISA kit (Shibayagi) according to the manufacturer's protocol. The amount of secreted insulin was normalized to the cellular protein content.

## Immunoblotting

Immunoblotting analysis was performed according to a standard procedure as previously described (*Ninagawa et al., 2011*). Chemiluminescence was generated using Western Blotting Luminol Reagent (Santa Cruz Biotechnology) and detected using an LAS-3000mini LuminoImage analyzer (Fuji Film). For detection of proinsulin and insulin, the pore size of PVDF membrane was changed from 0.45 to 0.2 μm (Amersham), and blocking and reaction with the anti-insulin monoclonal antibody was carried out in 5% BSA (Sigma-Aldrich) in PBS-0.1% Tween 20 buffer.

## Pulse-chase experiment and in vitro translation

Pulse-chase experiments using 9.8 Mbq per dish of EASY-TAG EXPRESS Protein labeling mix [$^{35}$S] (PerkinElmer) and subsequent immunoprecipitation using the anti-insulin monoclonal antibody and protein G-coupled Sepharose beads (GE Healthcare) were performed according to a published procedure (*Ninagawa et al., 2014*).

In vitro translation was performed using the TNT Quick Coupled Transcription/Translation Systems (Promega) and the EASY-TAG EXPRESS Protein labeling mix [$^{35}$S] (PerkinElmer) according to the manufacturer's instructions. Translated proteins were subjected to immunoprecipitation using the anti-insulin monoclonal antibody to separate them from $^{35}$S-methionine and $^{35}$S-cysteine. The immunoprecipitates were analyzed using SDS-PAGE (15 or 16% gel). Radiolabeled proteins were visualized using an FLA-3000G FluoroImage analyzer (Fuji Film).

## Immunofluorescence assay

For immunofluorescence analysis, untransfected MIN6 cells or cells transfected using Polyethyleneimine max were washed with PBS and fixed with 4% paraformaldehyde phosphate buffer (Nacalai Tesque) on ice for 4.5 min. Fixed cells were washed with PBS and permeabilized by incubation on ice for 4.5 min in PBS containing 0.2% Triton X-100. After incubation in PBS containing 3% fetal bovine serum and the primary antibody for 2 hr at room temperature, cells were incubated with secondary antibodies labeled with Alexa Fluors 488, 568, or 633 (Thermo Fisher Scientific) for 1 hr at room temperature. Images were acquired at room temperature at 100 × magnification using a DM IRE2 and confocal software (both from Leica).

## Mass spectrometric analysis

Nano-scale reversed-phase liquid chromatography coupled with tandem mass spectrometry (nanoLC/MS/MS) was performed with an Orbitrap Fusion Lumos mass spectrometer (Thermo Fisher Scientific), connected to a Thermo Ultimate 3000 RSLCnano pump and an HTC-PAL autosampler (CTC Analytics) equipped with a self-pulled analytical column (150 mm length ×100 μm i.d.) (*Ishihama et al., 2002*) packed with ReproSil-Pur C18-AQ materials (3 μm, Dr. Maisch GMBH). The mobile phases consisted of (A) 0.5% acetic acid and (B) 0.5% acetic acid and 80% acetonitrile. Peptides were eluted from the analytical column at a flow rate of 500 nl/min by altering the gradient: 5–10% B in 5 min, 10–40% B in 15 min, 40–100% B in 1 min and 100% for 4 min. The Orbitrap Fusion Lumos instrument was operated in the data-dependent mode with a full scan in the Orbitrap followed by MS/MS scans for 1.5 s using higher-energy collision dissociation (HCD). The applied voltage for ionization was 2.4 kV. The full scans were performed with a resolution of 120,000, a target value of $4 \times 10^5$ ions and a maximum injection time of 50 ms. The MS scan range was *m/z* 300–1500. The MS/MS scans were performed with a 15,000 resolution, a $5 \times 10^4$ target value and a 200 ms maximum injection time. Isolation window was set to 1.6 and normalized HCD collision energy was 30. Dynamic exclusion was applied for 20 s.

All raw datasets were analyzed and processed by MaxQuant (v1.6.2.3) (*Cox and Mann, 2008*). Default settings were employed. Search parameters included two missed cleavage sites and variable modifications such as methionine oxidation, protein N-terminal acetylation, cysteine carbamidomethyl and cysteine N-ethylmaleimide. The peptide mass tolerance was six ppm and the MS/MS tolerance was 20 ppm. Database search was performed with Andromeda (*Cox et al., 2011*) against the UniProt mouse database (downloaded on 2019–4) with common contaminants and enzyme

sequences. False discovery rate (FDR) was set to 1% at peptide spectrum match (PSM) level and at protein level. For protein quantification, total peak area of the peptides was used, and median peak area was calculated for each protein from three independent experiments. To compare protein abundance between the non-treated and the olanzapine-treated cells (see *Figure 8D*), we considered proteins that were quantified in all samples and replicates.

### ITC experiments

ITC measurements for the interaction between olanzapine and each PDI family protein were performed in buffer containing 50 mM HEPES-NaOH, pH 7.5, at 298 K and 750 r.p.m. For the preparation of olanzapine, 10 mg of olanzapine (32.0 mmol) was diluted in 500 µl of 50 mM HCl solution and freeze-dried using evaporator. 2.0 µl of the olanzapine solution (1.0 mM) was titrated into PDI family solutions (50 µM) at 180 s intervals after an initial 120 s delay. To minimize the effect of bubbles and imperfect solution filling of the syringe, the first titration was performed using 0.6 µl of solution in the syringe. The data were analyzed using MicroCal analysis (Malvern). The heats of dilution were subtracted from the raw binding data before analysis.

### Expression and purification of human proinsulin

Recombinant human proinsulin was expressed as inclusion bodies in *Escherichia coli* cells. Purification of proinsulin was carried out as previously described (*Okumura et al., 2011*). Briefly, inclusion bodies were treated with 100 mM Tris/HCl buffer, pH 8.0, containing 8 M urea and 10 mM DTT, and the solution was stood for 3 hr at 50℃. Reduced and denatured proinsulin was purified by RP-HPLC using Cosmosil 5C$_{18}$-AR-II (4.6 mm I.D. ×250 mm, Nacalai Tesque) monitored at 220 nm. Molecular mass of purified proinsulin was calculated using ProteinProspectors (http://prospector.ucsf.edu/prospector/mshome.htm) and its identity was confirmed by MALDI-TOF/MS. Purified proinsulin was lyophilized at −80℃ until used.

### Animals

Male BALB/c mice (8 weeks old) were purchased from Shimizu Laboratory Supplies. All mouse experiments were conducted under pathogen-free conditions and in line with Institutional Animal Care protocols approved by Kyoto University (Q 19–68). Administration of vehicle or olanzapine was conducted using a reusable oral gavage needle once a day. Olanzapine was dissolved in 1% acetic acid in saline buffered with 1M NaOH to a pH >5.5 (*McCormick et al., 2010*).

### Isolation of mouse pancreatic islets

To isolate islets from BALB/c mice, pancreas was inflated by injection of Hank's Balanced Salt Solution (HBSS) containing 0.15 mg/ml collagenase P (Roche Diagnostics) via the common bile duct. Distended pancreas was then excised and incubated at 37℃ for 18 min. After digested pancreas had dissociated, the tissue was washed with HBSS twice. The islets in the dissociated pancreatic tissue were purified on discontinuous gradients (1.110, 1.103, 1.096, and 1.070 g/ml) of OptiPrep (Axis-Shield) and ET Kyoto (ETK) solution (Otsuka Pharmaceutical). Isolated islets were cultured (37℃/5% $CO_2$/95% air humidified atmosphere) in RPMI1640 medium supplemented with 10% fetal bovine serum (FBS), 100 U/ml penicillin, and 100 µg/ml streptomycin.

## Acknowledgements

The authors declare no competing financial interests. We thank Kaoru Miyagawa for her technical and secretarial assistance, Dr. Akira Hattori (Kyoto University) and Ms. Masako Hirose (Malvern) for their help in the use of MicroCal Auto-iTC200, Dr. Jun-ichi Miyazaki for providing us MIN6 cells, Dr. Yuichi Tsuchiya (NAIST) and Dr. Masataka Kunii (Osaka University) for useful antibody information, and Ms. Nanae Fujimoto for her help in islet isolation. This work was financially supported in part by grants from the Ministry of Education, Culture, Sports, Science and Technology of Japan (18K06216 to S N, 19K06658 to T I, 17H01432 and 17H06419 to K M), the Joint Research Program of the Institute for Molecular and Cellular Regulation, Gunma University (#18021 to S N and S T), the Takeda Science Foundation (to MO) and Mochida Memorial Foundation for Medical and Pharmaceutical Research (to MO).

# Additional information

## Funding

| Funder | Grant reference number | Author |
|---|---|---|
| Ministry of Education, Culture, Sports, Science and Technology | 18K06216 | Satoshi Ninagawa |
| Ministry of Education, Culture, Sports, Science and Technology | 19K06658 | Tokiro Ishikawa |
| Ministry of Education, Culture, Sports, Science and Technology | 17H01432 | Kazutoshi Mori |
| Ministry of Education, Culture, Sports, Science and Technology | 17H06419 | Kazutoshi Mori |
| Gunma University | #18021 Joint Research Program of the Institute for Molecular and Cellular Regulation | Satoshi Ninagawa Seiji Torii |
| Takeda Science Foundation | | Masaki Okumura |
| Mochida Memorial Foundation for Medical and Pharmaceutical Research | | Masaki Okumura |

The funders had no role in study design, data collection and interpretation, or the decision to submit the work for publication.

## Author contributions

Satoshi Ninagawa, Investigation, Writing - original draft; Seiichiro Tada, Masaki Okumura, Kenta Inoguchi, Misaki Kinoshita, Shingo Kanemura, Koshi Imami, Hajime Umezawa, Tokiro Ishikawa, Yasushi Ishihama, Kenji Inaba, Takayuki Anazawa, Investigation; Robert B Mackin, Seiji Torii, Methodology; Takahiko Nagamine, Conceptualization; Kazutoshi Mori, Supervision, Funding acquisition, Writing - review and editing

## Author ORCIDs

Satoshi Ninagawa (iD) https://orcid.org/0000-0002-8005-4716
Seiichiro Tada (iD) https://orcid.org/0000-0001-8734-2270
Masaki Okumura (iD) https://orcid.org/0000-0001-8130-2470
Kenta Inoguchi (iD) https://orcid.org/0000-0003-0470-4251
Misaki Kinoshita (iD) https://orcid.org/0000-0001-5099-7572
Shingo Kanemura (iD) https://orcid.org/0000-0002-7012-8179
Koshi Imami (iD) http://orcid.org/0000-0002-7451-4982
Tokiro Ishikawa (iD) http://orcid.org/0000-0003-1718-6764
Robert B Mackin (iD) https://orcid.org/0000-0003-3297-9893
Seiji Torii (iD) https://orcid.org/0000-0002-7388-0331
Yasushi Ishihama (iD) http://orcid.org/0000-0001-7714-203X
Kenji Inaba (iD) http://orcid.org/0000-0001-8229-0467
Takayuki Anazawa (iD) https://orcid.org/0000-0002-7625-5750
Takahiko Nagamine (iD) http://orcid.org/0000-0002-0690-6271
Kazutoshi Mori (iD) https://orcid.org/0000-0001-7378-4019

## Ethics

Animal experimentation: All mouse experiments were conducted under pathogen-free conditions and in line with Institutional Animal Care protocols approved by Kyoto University (Q 19-68).

Decision letter and Author response
Decision letter https://doi.org/10.7554/eLife.60970.sa1
Author response https://doi.org/10.7554/eLife.60970.sa2

## Additional files

### Supplementary files

• Supplementary file 1. Data of mass spectrometric analysis Proteins quantified in all samples (see *Figure 8F*) are summarized.

• Supplementary file 2. Information of resources Nucleotide sequences of primers used are shown.

• Transparent reporting form

### Data availability

All data generated or analyzed during this study are included in the manuscript.

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
