## [Decision Letter]

**Acceptance summary:**

This analysis of the effect of the anti-psychotic olanzapine on misfolding and secretion of pro-insulin suggests a framework for understanding how such treatments might trigger rapid onset diabetes in patients. The authors elegantly show that olanzapine treatment alters the processing of pro-insulin such that an aggregation-prone intermediate accumulates, which is recalcitrant to secretion and thus accumulates in the endoplasmic reticulum. This has important implications for protein folding, secretion and ER stress in patients and provides a foundation for understanding the molecular basis by which this drug impacts protein folding.

**Decision letter after peer review:**

[Editors’ note: the authors submitted for reconsideration following the decision after peer review. What follows is the decision letter after the first round of review.]

Thank you for submitting your work entitled "Antipsychotic olanzapine targets proinsulin for endoplasmic reticulum-associated degradation by inducing its misfolding" for consideration by *eLife*. Your article has been reviewed by three peer reviewers, including Elizabeth A Miller as the Reviewing Editor and Reviewer #1, and the evaluation has been overseen by a Senior Editor.

Our decision has been reached after consultation between the reviewers. Based on these discussions and the individual reviews below, we regret to inform you that your work will not be considered further for publication in *eLife*.

Although there was enthusiasm for the research topic and the potential importance for human health, reviewers voiced significant concerns about the model systems. In particular, there was agreement that choice of the model is insufficient to support the claims with respect to disease/therapeutic relevance. Both reviewers 2 and 3 raised the problem of the lack of detection of insulin as a major concern, and had some suggestions regarding overcoming the different antibody specificity that might be helpful.

Reviewer #1:

Overall, this work provides strong evidence that OLA indirectly impairs insulin secretion by perturbing disulfide bond formation or rearrangement, thereby rendering pro-insulin trapped in the ER and subject to ERAD. This provides a nice explanation for physiological observations of diabetes induction in patients treated with this anti-psychotic medication and is therefore of general interest. However, whether the observed effects are direct or indirect remains to be clarified, which seems essential to warrant publication in *eLife*.

I have several specific concerns:

1) The quality of the immunofluorescence images is not particularly good.

2) A second disulfide-containing substrate should be investigated to show broad effects versus insulin-specific effects.

3) The activity of PDI should be assessed by proteomic analysis to test whether it is also forming mixed disulfides in a more promiscuous way, or if it is defective and thus the likely target of OLA.

Reviewer #2:

1) This paper wishes to address diabetes caused by olanzapine but there are no studies from drug-treated human or animal islets, which is a missed opportunity. Without islet work, the cell culture analysis really should be deeply insightful. In their previous publication they suggested that olanzapine blocks PERK action on eIF2α and "apoptosis was induced by olanzapine at the concentration of more than 50 μM" but neither PERK nor apoptosis appears in the current manuscript. In this manuscript, the authors show data suggesting that olanzapine does not bind proinsulin. So what is the primary drug target in β cells? It seems that most of what are measured in this manuscript are secondary effects of the drug without clarifying the primary effect. And, based on their previous paper, are the observed effects occurring in dying cells?

2) Figure 2B Figure legend says that the 26 ng of proinsulin run next to 2.9 micrograms of purified insulin, were subject to immunoblotting. The Results section concluded: "this antibody was more reactive with purified recombinant human proinsulin than purified recombinant human insulin". But no information is provided on whether the SDS-PAGE was performed under reducing or non-reducing conditions (the specific antibody also should be cited). It is essential that both conditions be shown in Figure 2B. Under reducing conditions, the band shown in the figure would not be insulin but one of the insulin chains split apart from insulin. If the immunoreactivity is much stronger with intact (non-reduced) insulin – then this might require significant re-interpretation of data within this manuscript – see below.

3) Figure 6C nonreducing gel shows three bands in all lanes in the region labeled as Proinsulin. If this anti-insulin antibody cannot see non-reduced insulin, then another immunoblot is required with an antibody that *can* recognize insulin. There is a simple question – do the untreated control β cells used in this study have any insulin? If not, is this a suitable β cell model? And how would the authors know whether olanzapine might affect the proteolytic processing from proinsulin to insulin? Conversely, if the antibody DOES see non-reduced insulin, are the authors entirely sure that the dark band in the control cells labeled Proinsulin (Figure 6C) is not actually Insulin? And if this is insulin, are we completely sure of which proteins are being recognized by immunofluorescence in the other figures?

4) The above discussion also raises the question of the physiological state of highly passaged Min6 cells. The authors cite a paper to suggest that their mouse β cells no longer express the mouse *Ins1* gene, which is clearly abnormal. Can they not test the relative expression level of mouse *Ins1* versus *Ins2* mRNA (such as from qPCR with known quantities of Ins1 and Ins2 cDNA standards)? This might help to clarify what the various bands represent.

5) Figure 7C and 8 – I'm not sure why the authors conclude that HMI-1 is a major ERAD substrate? In Figure 8 at 40, 80, and even 160 minutes of chase in OLA-treated cells, the biggest difference between with and without MG132 is the proinsulin monomer. How can the authors know which form is the ERAD substrate?

6) A shorter pulse labeling would demonstrate whether there is a precursor-product relationship to the two proinsulin bands in Figure 2E, and immunoprecipitates with anti mature-insulin should be shown to determine whether insulin is getting stored intracellularly. It is troubling if the authors cannot distinguish a "processed form" (label on Figure 2) from "partially degraded proinsulin" (subsection “Inhibition of insulin secretion and retention of proinsulin in the ER in olanzapine-treated cells”). Is this an artifact of the cell line?

Reviewer #3:

The manuscript submitted to *eLife* by Mori and colleagues reports on a potential molecular mechanism that may lead to metabolic disease in individuals taking second generation antipsychotics, such as olanzapine. A previous conjecture was that the drug increased appetite as a result of its effect on serotonin receptors, ultimately leading to diabetes. However, data in this paper potentially indicate why a more immediate effect on diabetic syndrome is evident, uncover why olanzapine led to mild UPR/PERK activation in a prior study, and identify a putative underlying cause of this more rapid effect.

The authors first report that olanzapine shuts-down the levels of secreted insulin in a murine insulinoma cell model in a dose-dependent manner, and that the drug blunts the level of secreted proinsulin in a pulse-chase analysis. The production/secretion of two other trafficked proteins were unaffected. Olanzapine also altered the steady-state localization of proinsulin, leading to its residence in the ER instead of in secretory granules. Consistent with these data as well as the effect of the drug on the UPR, ERAD-targeting of proinsulin increased under the conditions used in these experiments. In addition, the use of non-reducing gels indicated that olanzapine administration led to the presence of a high molecular weight species that was primarily composed of proinsulin dimers and trimers, and it was the "HMI-1" form that was mostly degraded via the ERAD pathway.

While the experiments support the idea that olanzapine triggers the ERAD of proinsulin, resulting in lower circulating levels and potentially in metabolic syndrome, there are numerous caveats of the study as it currently stands. These are outlined below. In addition, only one cell line was used and the relevance to human physiology is questionable for several reasons, which are also outlined below. Finally, the mechanism by which olanzapine affects the ER redox balance and leads to proinsulin misfolding is completely mysterious. In short, the manuscript only hints at the major conclusions claimed by the authors.

Essential revisions:

1) What is the justification for using MIN6 cells, other than the use of a convenient antibody for pulse-chase studies (Figure 2C)? These cells actually lose glucose-sensitive insulin secretion over time. There is also no justification of why 25 mM glucose was administered in the MIN6 cell experiments. This concentration is well beyond what might be encountered in vivo after feeding.

2) Similar to the concerns raised above, drugs were apparently administered for 14 hours for steady state studies and 4 hours for the pulse-chase. What are the pharmacodynamics of the drug in the body, and in patients what is the actually effective concentration that is "seen" by β cells? How much of the drug is protein-bound? This knowledge is vital as only small percentages of drug are "free" in sera. Moreover, the authors claim that patient sera harbors 0.016 – 0.24 μm olanzapine, which is still far below the doses used here. They also claim that this may be negated by patients with "lower catabolism". Which cytP450s are used to metabolize the drug, and is there any evidence that this varies in patients? Overall, it is not clear whether the concentrations and conditions used in these experiments is physiologically relevant.

3) The inability to detect insulin production in the pulse-chase is problematic (Figure 2D). However, a pseudo pulse-chase could be performed by brief starvation, medium removal/wash, followed by a glucose pulse, addition of CHX, and subsequent ELISA for insulin in the medium. Therefore, insulin secretion could be directly detected over time, which is essential to show.

4) Does an amino acid starvation period in a pulse-chase assay impact insulin production/secretion?

5) The fluorescence images are of low quality (Figure 4) and the reticular network of the ER is not apparent.

6) Does combined administration of BFA and ± olanzapine also show increased ERAD of proinsulin? This would prevent the confounding variable of having to take secretion efficiency into account. More trivially, why are the data in Figure 5D presented first?

7) While the authors focus on the "HMI" species, the drug also increases the amount of protein >38 kDa. In fact, this species is more prevalent than the HMI forms and may reflect aggregated species (Figure 6C). Indeed, these species may or may not also be targeted for ERAD (molecular weight markers were not included in Figure 8B).

8) The authors state that the experiment in Figure 8 was "more reliable" under the conditions of "these conditions". If so, then the magnitude of the olanzapine effect on the ERAD of proinsulin (±MG132) in Figure 8B, bottom, is quite modest, especially given some of the caveats noted above.

[Editors’ note: further revisions were suggested prior to acceptance, as described below.]

Thank you for submitting your article "The antipsychotic olanzapine-induced misfolding and ERAD of proinsulin accounts for atypical development of diabetes" for consideration by *eLife*. Your article has been reviewed by three peer reviewers, including Elizabeth A Miller as the Reviewing Editor and Reviewer #1, and the evaluation has been overseen by Vivek Malhotra as the Senior Editor.

The reviewers have discussed the reviews with one another and the Reviewing Editor has drafted this decision to help you prepare a revised submission. After discussion among the reviewers and senior editor, we agreed that your study provides some thought-provoking new insight into the molecular mechanisms by which diabetes may arise rapidly after onset of patient treatment with olanzapine. That said, there was agreement among reviewers that some of the claims should be toned down, and you should state more clearly the limitations and caveats. Most importantly, the focus on ERAD (including in the title) was considered unwarranted. There was concern about the extent to which ERAD can account for the observed phenotypes, as well as the problem that MG132 treatment does not rescue secretion even if degradation is blocked. Therefore, ERAD is unlikely to explain insulin secretion defect, although reduced insulin secretion could partially explain the ERAD (this was the point of the BFA experiment suggested by Reviewer 3 in the original evaluation). ERAD should thus be appropriately de-emphasized as a driving mechanism. The other over-riding concern was about the concentrations of olanzapine that might accrue in patient cells. Of course, such measures are difficult to make, but by our estimates the effective concentration in your in vitro proinsulin folding experiment is at more than an order of magnitude higher than could possibly be achieved in patient tissue. Thus, the experiment shown in Figure 8C is particularly problematic, where a very high level of drug is required to see a quite subtle in vitro effect. The reviewers agreed that these data muddy the waters and should be removed. Thus, the ultimate conclusion is that although the mechanism of action remains unclear, your findings still provide insight into proinsulin misfolding as a potential side-effect of olanzapine treatment. You should be more careful in claiming a direct mechanism since that is not supported by the data shown.

Reviewer #1:

In this revised version, the authors have added another antibody to establish the secretion of pro-insulin and insulin in their cell model, but many of the key experiments are all still only performed with the original antibody that doesn't recognise insulin. Thus, the primary concern (expressed by both other reviewers) that the precursor-product relationship cannot be established, has not been fully resolved. My interpretation of the data is that OLA treatment prevents the production of the P' ("processed proinsulin") form, but it was hard for me to know exactly what the P' protein corresponds to. One interpretation is that P' is aggregation prone (to yield the HMP products) and subject to ERAD. So, I suspect that most of the conclusions are correct but greater clarity is needed regarding the P' protein (what it is and how it changes with OLA treatment). This should therefore be quantified in all of the pulse-chase experiments presented (it is currently missing from many). The full precursor-product analysis requested in the original critique would go a long way to clarifying this.

Reviewer #2:

In the revised manuscript submitted to *eLife*, Mori and colleagues have added new data and models, and have addressed many of the comments that previously precluded publication. A better characterization of the antibodies, the inclusion of data with islets and other (unaffected) substrates, an effect with lower doses of olanzapine, and improved fluorescence images further support the authors' earlier conclusions. However, there is still the question of whether the effect of the drug is direct or indirect. An attempt to show interaction with PDIs, which was a long-shot, provided negative data (also see below), and even at the lower concentration of 10 uM, this value is still ~40x higher than serum levels in humans taking the drug. It is likely that the intracellular concentration in patients is even lower, given considerations regarding logP and the free vs bound population of the drug. Two reviewers also expressed doubt that ERAD targeting was a major contributor to reduced levels of pro-insulin. With the exception of a modest effect of MG132, there are no further supporting data. Therefore, the title and statements such as "indicating that a part of newly synthesized proinsulin is constitutively subjected to ERAD" are premature. In fact, it was surprising that a putative effect of MG132 in the islets was not examined, especially since "the level of proinsulin was markedly decreased 20 hours after incubation with 5 μm and 10 μm olanzapine". Moreover, a previous comment to determine whether enhanced ERAD targeting might be apparent when ER protein export is inhibited (for, example, with BFA or dominant negative SAR1) was completely misinterpreted by the authors in the rebuttal, who replied "We have not used BFA in this manuscript".

Essential revisions:

The experiments using ITC are problematic. 1 mM olanzapine was used, so any acquired data may be irrelevant to the results in cells or rodents. Furthermore, the in vitro effect of 1 mM on the refolding of denatured/reduced proinsulin lacks a key control. Does 1 mM risperidone also have the same, subtle effect?

In the experiments using mice in which there is an increase in the amount of HMP-1, there is similarly no control for olanzapine. Does risperidone, or an (inactive) olanzapine derivative, have the same effect?

Mass spec data (coverage, number of peptides) are not shown.

In subsection “Inhibition of insulin secretion in olanzapine-treated MIN6 cells”, risperidone is used without any mention of what this compound is or why it was chosen as a control.

In subsection “Retention of proinsulin in the ER in olanzapine-treated MIN6 cells”, a previous comment in the review with regard to AiPI was ignored: was it expected that the MIN6 cells express this protein? (Another reviewer commented on the prolonged passage of these cells and whether they truly reflect islet biology.) More mundane, the statement "Intracellular localization of A1PI was not affected…" is not clear. Do the authors mean that secretion was unaffected? Finally, what is the significance of using the anti-KDEL antibody. This is mentioned in the absence of any context.

The labeling of lanes/times in Figure 2E and Figure 3B needs to be fixed, and the labeling/spacing at the top of Figure 10A-C is odd.

In the Discussion section, the authors state, "olanzapine produced aberrantly disulfide-bonded proinsulin….leading to β-cell dysfunction". Are there any data to support this? The authors later state that the drug did not induce apoptosis, which appears to be in conflict with this earlier claim.

Reviewer #3:

The manuscript is much improved but Figure 8C and the conclusions associated with that panel are seriously flawed. Were it not for this panel I believe the paper could be acceptable for publication. There is also a small matter of Figure 9D.

Problems with Figure 8C:

1) 100 μm OLA kills β cells, but here the authors are adding 10 times more than this to try to get an effect in an in vitro folding assay. No conclusion can be reached from this experiment, which puts the text of the manuscript in this part at risk of not being supported by real data. The interpretation that the drug acts directly on proinsulin is almost certainly wrong – so why imply such a thing when it is totally unnecessary to the paper?

2) The starting material in this experiment seems to have a molecular mass that seems to be too big to be monomeric proinsulin.

Figure 9D:

Don't the authors want to comment on the fact that the monomeric proinsulin in the nonreducing gel is also protected by MG132? It is clearly shown on the gel and the quantification in the graph below. Why would the authors not make this point?

---

## [Author Response]

[Editors’ note: the authors resubmitted a revised version of the paper for consideration. What follows is the authors’ response to the first round of review.]

Reviewer #1:Overall, this work provides strong evidence that OLA indirectly impairs insulin secretion by perturbing disulfide bond formation or rearrangement, thereby rendering pro-insulin trapped in the ER and subject to ERAD. This provides a nice explanation for physiological observations of diabetes induction in patients treated with this anti-psychotic medication and is therefore of general interest. However, whether the observed effects are direct or indirect remains to be clarified, which seems essential to warrant publication in eLife.I have several specific concerns:1) The quality of the immunofluorescence images is not particularly good.

It appears very difficult to obtain clear immunofluorescence image of β cells. Please note that the right figure shows immunofluorescence analysis of INS1 cells (Figure 1C of Arunagiri et al., 2019).

2) A second disulfide-containing substrate should be investigated to show broad effects versus insulin-specific effects.

We have shown that olanzapine also did not significantly affect the maturation of hemagglutinin from high mannose-type to complex type (Figure 5E), which requires correct disulfide bond formation for folding (Segal et al., 1992).

3) The activity of PDI should be assessed by proteomic analysis to test whether it is also forming mixed disulfides in a more promiscuous way, or if it is defective and thus the likely target of OLA.

We did not observe any changes in disulfide-bonded status of PDI in mouse islets after treatment with olanzapine (Figure 10B). Although we could not detect direct interaction between olanzapine and oxidoreductases in the ER by ITC (Figure 8A), we have found that olanzapine inhibited oxidative folding of purified proinsulin, leading to aberrant formation of intermolecular disulfide bonds (Figure 8C).

Reviewer #2:1) This paper wishes to address diabetes caused by olanzapine but there are no studies from drug-treated human or animal islets, which is a missed opportunity. Without islet work, the cell culture analysis really should be deeply insightful.

We have shown that olanzapine-induced misfolding of proinsulin also occurred when mouse islets were treated with olanzapine (Figure 10A-C). Furthermore, misfolding of proinsulin was induced in mouse islets after daily oral administration of 3 mg/kg/day olanzapine (Figure 10D-E).

In their previous publication they suggested that olanzapine blocks PERK action on eIF2α and "apoptosis was induced by olanzapine at the concentration of more than 50 μM" but neither PERK nor apoptosis appears in the current manuscript. In this manuscript, the authors show data suggesting that olanzapine does not bind proinsulin. So what is the primary drug target in β cells? It seems that most of what are measured in this manuscript are secondary effects of the drug without clarifying the primary effect. And, based on their previous paper, are the observed effects occurring in dying cells?

We have found that olanzapine inhibited oxidative folding of purified proinsulin, leading to aberrant formation of intermolecular disulfide bonds (Figure 8C).

We have described that “Inhibition of insulin secretion as well as proinsulin secretion (Figures 1C and Figure 4B) and retention of misfolded proinsulin in the ER (Figure 6D) was observed in MIN6 cells treated with 10 µM olanzapine (Discussion section)” and that “It should be noted that olanzapine-induced apoptosis was not previously observed in MIN6 cells or in the hamster pancreatic β−cell line HIT-T15 treated with 10 µM olanzapine (Ozasa et al., 2013), indicating that olanzapine-induced misfolding of proinsulin is an early event which evokes ER stress in pancreatic β cells (Discussion section)".

2) Figure 2B Figure legend says that the 26 ng of proinsulin run next to 2.9 micrograms of purified insulin, were subject to immunoblotting. The Results section concluded: "this antibody was more reactive with purified recombinant human proinsulin than purified recombinant human insulin". But no information is provided on whether the SDS-PAGE was performed under reducing or non-reducing conditions (the specific antibody also should be cited). It is essential that both conditions be shown in Figure 2B. Under reducing conditions, the band shown in the figure would not be insulin but one of the insulin chains split apart from insulin. If the immunoreactivity is much stronger with intact (non-reduced) insulin – then this might require significant re-interpretation of data within this manuscript – see below.

We have improved Figure 2 and described that “It should be noted that #8138 was 800-fold more reactive with purified recombinant human proinsulin than purified recombinant human insulin (insulin B chain) under reducing conditions (Figure 2C) ( – subsection “Retention of proinsulin in the ER in olanzapine-treated MIN6 cells”)”.

3) Figure 6C nonreducing gel shows three bands in all lanes in the region labeled as Proinsulin. If this anti-insulin antibody cannot see non-reduced insulin, then another immunoblot is required with an antibody that can recognize insulin. There is a simple question – do the untreated control β cells used in this study have any insulin? If not, is this a suitable β cell model? And how would the authors know whether olanzapine might affect the proteolytic processing from proinsulin to insulin? Conversely, if the antibody DOES see non-reduced insulin, are the authors entirely sure that the dark band in the control cells labeled Proinsulin (Figure 6C) is not actually Insulin? And if this is insulin, are we completely sure of which proteins are being recognized by immunofluorescence in the other figures?4) The above discussion also raises the question of the physiological state of highly passaged Min6 cells. The authors cite a paper to suggest that their mouse β cells no longer express the mouse Ins1 gene, which is clearly abnormal. Can they not test the relative expression level of mouse Ins1 versus Ins2 mRNA (such as from qPCR with known quantities of Ins1 and Ins2 cDNA standards)? This might help to clarify what the various bands represent.

We have employed mouse monoclonal antibody I2018 which immunoprecipitated mature proinsulin 2 and insulin 2 in addition to #8138 previously used, and have shown that MIN6 cells secrete both mature proinsulin and insulin (non-reducing conditions, Figure 2E).

5) Figure 7C and 8 – I'm not sure why the authors conclude that HMI-1 is a major ERAD substrate? In Figure 8 at 40, 80, and even 160 minutes of chase in OLA-treated cells, the biggest difference between with and without MG132 is the proinsulin monomer. How can the authors know which form is the ERAD substrate?

We have described that “Treatment of MIN6 cells with the proteasome inhibitor MG132 alone increased the levels of intracellular and extracellular proinsulin significantly (Figure 4C and 4D left), indicating that a part of newly synthesized proinsulin is constitutively subjected to ERAD in MIN6 cells (subsection “Induction of proinsulin misfolding in olanzapine-treated MIN6 cells”)” and that “Critically, analysis under non-reducing conditions of immunoprecipitates obtained with #8138 revealed that MG132 stabilized proinsulin, HMP-1, and HMP-2 in olanzapine-treated MIN6 cells (Figure 9D, lower part). These results strongly suggest that HMP-1 and HMP-2 underwent ERAD (subsection “ERAD of misfolded proinsulin in olanzapine-treated MIN6 cells and mouse islets”)”.

6) A shorter pulse labeling would demonstrate whether there is a precursor-product relationship to the two proinsulin bands in Figure 2E, and immunoprecipitates with anti mature-insulin should be shown to determine whether insulin is getting stored intracellularly. It is troubling if the authors cannot distinguish a "processed form" (label on Figure 2) from "partially degraded proinsulin" (subsection “Inhibition of insulin secretion and retention of proinsulin in the ER in olanzapine-treated cells”). Is this an artifact of the cell line?

We have shown that proinsulin (P) was accumulated and instead processed proinsulin (P’), mature prinsulin and insulin were decreased in olanzapine-treated MIN6 cells (Figure 3), indicating that olanzapine blocked maturation of proinsulin and accordingly inhibited secretion of insulin (subsection “Retention of proinsulin in the ER in olanzapine-treated MIN6 cells”).

Reviewer #3:[…]Essential revisions:1) What is the justification for using MIN6 cells, other than the use of a convenient antibody for pulse-chase studies (Figure 2C)? These cells actually lose glucose-sensitive insulin secretion over time. There is also no justification of why 25 mM glucose was administered in the MIN6 cell experiments. This concentration is well beyond what might be encountered in vivo after feeding.2) Similar to the concerns raised above, drugs were apparently administered for 14 hours for steady state studies and 4 hours for the pulse-chase. What are the pharmacodynamics of the drug in the body, and in patients what is the actually effective concentration that is "seen" by β cells? How much of the drug is protein-bound? This knowledge is vital as only small percentages of drug are "free" in sera. Moreover, the authors claim that patient sera harbors 0.016 – 0.24 μm olanzapine, which is still far below the doses used here.

We have shown that proinsulin misfolding occurred in MIN6 cells treated with 10 µM and 50 µM olanzapine (Figure 3 and Figure 4) and that HMP-1 and even higher molecular weight forms of proinsulin were produced when isolated mouse islets were incubated for 4 hours with 10 µM and 50 µM olanzapine (Figure 10A) and for 20 hours with 5 µM and 10 µM olanzapine (Figure 10B).

They also claim that this may be negated by patients with "lower catabolism". Which cytP450s are used to metabolize the drug, and is there any evidence that this varies in patients? Overall, it is not clear whether the concentrations and conditions used in these experiments is physiologically relevant.

We have described that “This rapidity and broad extensiveness of olanzapine’s action on the folding status of proinsulin in mouse islets suggests that β-cell dysfunction may infrequently dominate and diabetes might develop without weight gain or insulin resistance in the minority of patients with potentially lower catabolism of olanzapine or by other unidentified factors. For example, it is known that olanzapine is catabolized by the action of cytochrome p450 (CPY) 1A2 and polymorphic CPY2D6 in the liver and that the induction of CPY1A2 by smoking significantly diminishes the concentration of olanzapine in plasma (Carrillo et al., 2003); as hospitalized patients cannot smoke, the concentration of olanzapine may be elevated in plasma and possibly in β cells of ex-smokers. The clearance of olanzapine also appears to vary by race, sex and age (Meibohm et al., 2002)” (Discussion section).

3) The inability to detect insulin production in the pulse-chase is problematic (Figure 2D). However, a pseudo pulse-chase could be performed by brief starvation, medium removal/wash, followed by a glucose pulse, addition of CHX, and subsequent ELISA for insulin in the medium. Therefore, insulin secretion could be directly detected over time, which is essential to show.4) Does an amino acid starvation period in a pulse-chase assay impact insulin production/secretion?

We have employed mouse monoclonal antibody I2018 which immunoprecipitated mature proinsulin 2 and insulin 2 in addition to #8138 previously used, and have shown that MIN6 cells secrete both mature proinsulin and insulin (non-reducing conditions, Figure 2E).

5) The fluorescence images are of low quality (Figure 4) and the reticular network of the ER is not apparent.

Please see our response to the comment 1 of the reviewer #1.

6) Does combined administration of BFA and ± olanzapine also show increased ERAD of proinsulin? This would prevent the confounding variable of having to take secretion efficiency into account. More trivially, why are the data in Figure 5D presented first?

We have not used BFA data in this manuscript.

7) While the authors focus on the "HMI" species, the drug also increases the amount of protein >38 kDa. In fact, this species is more prevalent than the HMI forms and may reflect aggregated species (Figure 6C). Indeed, these species may or may not also be targeted for ERAD (molecular weight markers were not included in Figure 8B).

In MIN6 cells, proteins >38 kDa were observed in even control cells, and HMP-1 and HMP-2 were induced by olanzapine treatment (Figure 9). In mouse islets, proteins >38 kDa were also induced by olanzapine treatment (Figure 10). Most importantly, we have succeeded in showing the decrease in the amount of proinsulin in mouse islets after treatment with olanzapine (Figure 10).

8) The authors state that the experiment in Figure 8 was "more reliable" under the conditions of "these conditions". If so, then the magnitude of the olanzapine effect on the ERAD of proinsulin (±MG132) in Figure 8B, bottom, is quite modest, especially given some of the caveats noted above.

We have replaced old Figure 8 with new Figure 9.

[Editors’ note: what follows is the authors’ response to the second round of review.]

The reviewers have discussed the reviews with one another and the Reviewing Editor has drafted this decision to help you prepare a revised submission. After discussion among the reviewers and senior editor, we agreed that your study provides some thought-provoking new insight into the molecular mechanisms by which diabetes may arise rapidly after onset of patient treatment with olanzapine. That said, there was agreement among reviewers that some of the claims should be toned down, and you should state more clearly the limitations and caveats. Most importantly, the focus on ERAD (including in the title) was considered unwarranted. There was concern about the extent to which ERAD can account for the observed phenotypes, as well as the problem that MG132 treatment does not rescue secretion even if degradation is blocked. Therefore, ERAD is unlikely to explain insulin secretion defect, although reduced insulin secretion could partially explain the ERAD (this was the point of the BFA experiment suggested by Reviewer 3 in the original evaluation). ERAD should thus be appropriately de-emphasized as a driving mechanism.

We have removed the word “ERAD” from the title and modified the main text based on our claim that olanzapine-induced misfolding of proinsulin and retention of misfolded proinsulin in the ER account for blockage of insulin secretion.

The other over-riding concern was about the concentrations of olanzapine that might accrue in patient cells. Of course, such measures are difficult to make, but by our estimates the effective concentration in your in vitro proinsulin folding experiment is at more than an order of magnitude higher than could possibly be achieved in patient tissue. Thus, the experiment shown in Figure 8C is particularly problematic, where a very high level of drug is required to see a quite subtle in vitro effect. The reviewers agreed that these data muddy the waters and should be removed. Thus, the ultimate conclusion is that although the mechanism of action remains unclear, your findings still provide insight into proinsulin misfolding as a potential side-effect of olanzapine treatment. You should be more careful in claiming a direct mechanism since that is not supported by the data shown.

We have removed the data in the previous Figure 8C and added that “although direct targets of olanzapine remain undetermined” to the Abstract and “Thus, direct targets of olanzapine remain to be determined” to the final paragraph of subsection “Induction of proinsulin misfolding in olanzapine-treated MIN6 cells”.

Reviewer #1:In this revised version, the authors have added another antibody to establish the secretion of pro-insulin and insulin in their cell model, but many of the key experiments are all still only performed with the original antibody that doesn't recognise insulin. Thus, the primary concern (expressed by both other reviewers) that the precursor-product relationship cannot be established, has not been fully resolved. My interpretation of the data is that OLA treatment prevents the production of the P' ("processed proinsulin") form, but it was hard for me to know exactly what the P' protein corresponds to. One interpretation is that P' is aggregation prone (to yield the HMP products) and subject to ERAD. So, I suspect that most of the conclusions are correct but greater clarity is needed regarding the P' protein (what it is and how it changes with OLA treatment). This should therefore be quantified in all of the pulse-chase experiments presented (it is currently missing from many). The full precursor-product analysis requested in the original critique would go a long way to clarifying this.

The precursor-product relationship for proinsulin 2 and insulin 2 was already and clearly shown by pulse (20 minutes)-chase experiment using I2018 (Figure 3E and Figure 4B, lanes 15-18). We have added that “This also demonstrated the precursor-product relationship for proinsulin 2 and insulin 2 in MIN6 cells” (subsection “Retention of proinsulin in the ER in olanzapine-treated MIN6 cells”). We have also shown that olanzapine did not alter the localization of insulin (Figure 8A and 8B), allowing us to focus on the behavior of proinsulin in olanzapine-treated cells.

We have now shown the precursor-product relationship for proinsulin 2 and processed proinsulin 2 by conducting shorter pulse (3 min)-chase experiments using #8138 (Figure 3C).

We have noted that “The lower band was considered to be processed proinsulin 2, such as the B chain connected to the C peptide but not to the A chain as observed in isolated rat islets (Harding et al., 2012), or somewhat cleaved proinsulin, because only proinsulin 2 was detected after a shorter pulse (3minutes) and processed proinsulin 2 appeared after the 30-minutes chase (Figure 3C)” (subsection “Retention of proinsulin in the ER in olanzapine-treated MIN6 cells”).

We believe that processed proinsulin is unlikely to be aggregation prone (to yield the HMP products) because it was secreted (Figure 3B) and that it is likely to be produced around the Golgi apparatus because the level of processed proinsulin was markedly decreased by treatment with brefeldin A (Figure 3D). Our explanation for why OLA treatment prevents the production of processed proinsulin is that OLA treatment induced misfolding of proinsulun and retention of misfolded proinsulin in the ER.

We have noted that “Processed proinsulin 2 is likely to be produced after moving out from the ER, because its level was markedly decreased by treatment with brefeldin A, an inhibitor of anterograde transport from the ER to the Golgi apparatus (Figure 3D)” (subsection “Retention of proinsulin in the ER in olanzapine-treated MIN6 cells”).

Reviewer #2:In the revised manuscript submitted to eLife, Mori and colleagues have added new data and models, and have addressed many of the comments that previously precluded publication. A better characterization of the antibodies, the inclusion of data with islets and other (unaffected) substrates, an effect with lower doses of olanzapine, and improved fluorescence images further support the authors' earlier conclusions. However, there is still the question of whether the effect of the drug is direct or indirect. An attempt to show interaction with PDIs, which was a long-shot, provided negative data (also see below), and even at the lower concentration of 10 uM, this value is still ~40x higher than serum levels in humans taking the drug. It is likely that the intracellular concentration in patients is even lower, given considerations regarding logP and the free vs bound population of the drug.

We would draw your attention to the finding that daily oral administration of olanzapine produced HMP-1 and even higher molecular weight forms of proinsulin in mouse islets (Figure 10E-F), albeit that we could not determine the concentration of olanzapine in serum.

Two reviewers also expressed doubt that ERAD targeting was a major contributor to reduced levels of pro-insulin. With the exception of a modest effect of MG132, there are no further supporting data. Therefore, the title and statements such as "indicating that a part of newly synthesized proinsulin is constitutively subjected to ERAD" are premature.

We have removed the word “ERAD” from the title.

We consider that the increase in the level of intracellular proinsulin in MIN6 cells treated with both olanzapine and MG132 over that in MIN6 cells treated with olanzapine alone is marked (Figure 5D, right top panel). Instead of toning this down, we have added that “Nonetheless, ERAD of retained proinsulin alone cannot explain olanzapine-induced blockage of proinsulin secretion, because proinsulin secretion was still blocked in MIN6 cells treated with both olanzapine and MG132 (Figure 5D, right bottom panel), in contrary to the case with control cells (Figure 5D, left bottom panel)” (subsection “Induction of proinsulin misfolding in olanzapine-treated MIN6 cells”).

In fact, it was surprising that a putative effect of MG132 in the islets was not examined, especially since "the level of proinsulin was markedly decreased 20 hours after incubation with 5 μm and 10 μm olanzapine".

We have now shown this effect and added that “Reducing SDS-PAGE analysis followed by immunoblotting showed that the level of proinsulin was decreased 4 hours after incubation with 50 µM olanzapine (Figure 10A) and that this decrease was blocked by the addition of 30 µM MG132 from 1 hour earlier (total 5 hours) (Figure 10D)” (subsection “ERAD of misfolded proinsulin in olanzapine-treated MIN6 cells and mouse islets”).

We have avoided the use of MG132 for 20 hours, considering its toxicity.

Moreover, a previous comment to determine whether enhanced ERAD targeting might be apparent when ER protein export is inhibited (for, example, with BFA or dominant negative SAR1) was completely misinterpreted by the authors in the rebuttal, who replied "We have not used BFA in this manuscript".

We apologize for our insufficient explanation. We tried to examine the effect of combined administration of brefeldin A and ± olanzapine on ERAD of proinsulin. To this end, we must treat MIN6 cells with (1) none, (2) olanzapine, (3) brefeldin A, (4) olanzapine and brefeldin, and (5) olanzapine, brefeldin A and MG132. During this experiment, we found that brefeldin A is quite toxic to MIN6 cells. In most cases, when MIN6 cells were treated with brefeldin A for longer periods, a lower amount of radioisotope was incorporated into proteins, as shown below. We thought that it is difficult to obtain meaningful results with this experiment.

Essential revisions:The experiments using ITC are problematic. 1 mM olanzapine was used, so any acquired data may be irrelevant to the results in cells or rodents. Furthermore, the in vitro effect of 1 mM on the refolding of denatured/reduced proinsulin lacks a key control. Does 1 mM risperidone also have the same, subtle effect?

We have added that “although direct targets of olanzapine remain undetermined” to the Abstract and “Thus, direct targets of olanzapine remain to be determined” to subsection “Induction of proinsulin misfolding in olanzapine-treated MIN6 cells”. We have removed the data of the refolding of denatured/reduced proinsulin.

In the experiments using mice in which there is an increase in the amount of HMP-1, there is similarly no control for olanzapine. Does risperidone, or an (inactive) olanzapine derivative, have the same effect?

Risperidone did not increase the amount of HMP-1 over control (Figure 10C).

Mass spec data (coverage, number of peptides) are not shown.

The data are supplied as Table 1.

In subsection “Inhibition of insulin secretion in olanzapine-treated MIN6 cells”, risperidone is used without any mention of what this compound is or why it was chosen as a control.

We have described this as “However, it carries a higher risk of diabetes than other SGAs such as risperidone” (Introduction).

In subsection “Retention of proinsulin in the ER in olanzapine-treated MIN6 cells”, a previous comment in the review with regard to AiPI was ignored: was it expected that the MIN6 cells express this protein? (Another reviewer commented on the prolonged passage of these cells and whether they truly reflect islet biology.) More mundane, the statement "Intracellular localization of A1PI was not affected…" is not clear. Do the authors mean that secretion was unaffected? Finally, what is the significance of using the anti-KDEL antibody. This is mentioned in the absence of any context.

A1AT was expressed in MIN6 cells by transfection. We have changed this to “The antiKDEL antibody stained the ER by recognizing the major ER chaperones BiP/GRP78 and GRP94, and A1PI was widely distributed from the ER the Golgi apparatus before and after olanzapine treatment” (Figure 6C and 6D)” (subsection “Retention of proinsulin in the ER in olanzapine-treated MIN6 cells”).

The labeling of lanes/times in Figure 2E and 3B needs to be fixed, and the labeling/spacing at the top of Figure 10A-C is odd.

Please tell us more specifically what is odd and how it should be fixed.

In the Discussion section, the authors state, "olanzapine produced aberrantly disulfide-bonded proinsulin….leading to β-cell dysfunction". Are there any data to support this? The authors later state that the drug did not induce apoptosis, which appears to be in conflict with this earlier claim.

Dysfunction meant little insulin secretion, but we have removed “leading to b-cell dysfunction (Figure 10G)” (Discussion section).

Reviewer #3:The manuscript is much improved but Figure 8C and the conclusions associated with that panel are seriously flawed. Were it not for this panel I believe the paper could be acceptable for publication. There is also a small matter of Figure 9D.Problems with Figure 8C:1) 100 μm OLA kills β cells, but here the authors are adding 10 times more than this to try to get an effect in an in vitro folding assay. No conclusion can be reached from this experiment, which puts the text of the manuscript in this part at risk of not being supported by real data. The interpretation that the drug acts directly on proinsulin is almost certainly wrong – so why imply such a thing when it is totally unnecessary to the paper?

We have removed the data in the previous Figure 8C.

2) The starting material in this experiment seems to have a molecular mass that seems to be too big to be monomeric proinsulin.

This was modified with AMS to stop the reaction, as was described previously in the legend.

Figure 9D:Don't the authors want to comment on the fact that the monomeric proinsulin in the nonreducing gel is also protected by MG132? It is clearly shown on the gel and the quantification in the graph below. Why would the authors not make this point?

We have described this as “MG132 stabilized proinsulin monomer, HMP-1, and HMP-2 in olanzapine-treated MIN6 cells (Figure 9D, lower part). These results strongly suggest that proinsulin monomer, HMP-1, and HMP-2 with aberrant disulfide bonds underwent ERAD (see Figure 10G)” (subsection “ERAD of misfolded proinsulin in olanzapine-treated MIN6 cells and mouse islets”).